# Twenty-first-century demographic and social inequalities of heat-related deaths in Brazilian urban areas

Djacinto Monteiro dos Santos[1]*, Renata Libonati[1,2,3]*, Beatriz N. Garcia[1¤], João L. Geirinhas[2], Barbara Bresani Salvi[4], Eliane Lima e Silva[5,6], Julia A. Rodrigues[1], Leonardo F. Peres[1], Ana Russo[2], Renata Gracie[7], Helen Gurgel[5,6], Ricardo M. Trigo[1,2]

1 Departamento de Meteorologia, Universidade Federal do Rio de Janeiro, Rio de Janeiro, Brazil, 2 Universidade de Lisboa, Faculdade de Ciências, Instituto Dom Luiz, Lisbon, Portugal, 3 Forest Research Centre, School of Agriculture, University of Lisbon, Lisbon, Portugal, 4 Escola Nacional de Saúde Pública Sergio Arouca - ENSP/ Fiocruz - Programa de Pós Graduação em Saúde Pública e Meio Ambiente, 5 Departamento de Geografia, Universidade de Brasilia, Distrito Federal, Brazil, 6 LMI Sentinela, International Joint Laboratory "Sentinela" (Fiocruz, UnB, IRD), Distrito Federal, Brazil, 7 Instituto de Comunicação e Informação Científica e Tecnológica em Saúde - ICICT/Fiocruz Rio de Janeiro, Rio de Janeiro, Brazil

¤ Current address: Energy Planning Program, Graduate School of Engineering, Federal University of Rio de Janeiro (COPPE/UFRJ), Rio de Janeiro, Brazil
* renata.libonati@igeo.ufrj.br (RL); santos.djacinto@gmail.com (DMS)

**Data Availability Statement:** The data underlying the results presented in the study are available from: Daily maximum and minimum surface air

## Abstract

Population exposure to heat waves (HWs) is increasing worldwide due to climate change, significantly affecting society, including public health. Despite its significant vulnerabilities and limited adaptation resources to rising temperatures, South America, particularly Brazil, lacks research on the health impacts of temperature extremes, especially on the role played by socioeconomic factors in the risk of heat-related illness. Here, we present a comprehensive analysis of the effects of HWs on mortality rates in the 14 most populous urban areas, comprising approximately 35% of the country's population. Excess mortality during HWs was estimated through the observed-to-expected ratio (O/E) for total deaths during the events identified. Moreover, the interplay of intersectionality and vulnerability to heat considering demographics and socioeconomic heterogeneities, using gender, age, race, and educational level as proxies, as well as the leading causes of heat-related excess death, were assessed. A significant increase in the frequency was observed from the 1970s (0–3 HWs year⁻¹) to the 2010s (3–11 HWs year⁻¹), with higher tendencies in the northern, northeastern, and central-western regions. Over the 2000–2018 period, 48,075 (40,448–55,279) excessive deaths were attributed to the growing number of HWs (>20 times the number of landslides-related deaths for the same period). Nevertheless, our event-based surveillance analysis did not detect the HW-mortality nexus, reinforcing that extreme heat events are a neglected disaster in Brazil. Among the leading causes of death, diseases of the circulatory and respiratory systems and neoplasms were the most frequent. Critical regional differences were observed, which can be linked to the sharp North-South inequalities in terms of socioeconomic and health indicators, such as life expectancy. Higher heat-related excess mortality was observed for low-educational level people, blacks and browns, older adults,

temperature data (1970-2020) were provided by the Brazilian National Institute of Meteorology (INMET, https://portal.inmet.gov.br/, accessed 04 October 2021) and the Brazilian Institute of Air Space Control (ICEA, https://www.icea.decea.mil.br/, accessed 06 April 2022). Daily mortality data from the Brazilian Health Informatics Department (DATASUS) for the 2000–2018 period were provided by the Data Science Platform applied to Health (PCDaS) of the Oswaldo Cruz Foundation (https://pcdas.icict.fiocruz.br/, accessed 04 October 2021).

**Funding:** D.M.S. acknowledges the support of FIOCRUZ [grant VPPCB-003-FIO-19], FAPERJ [grant E-26/205.890/2022]. RL was supported by FAPERJ [grant E-26/200.329/2023 and E-26/210.078/2023] and CNPQ [grant 311487/2021-1]. A.R. and R.M.T. were supported by Fundação para a Ciência e a Tecnologia, I.P./ MCTES through national funds (PIDDAC)" –UIDB/50019/2020 and also by Project ROADMAP (JPIOCEANS/0001/2019). B.N.G. was supported by CNPQ [grant 161075/2021-5]. J.L.G. acknowledges the support of FCT (Fundação para a Ciência e Tecnologia) for the PhD Grant 2020.05198.BD. HG was supported by CNPQ [grant 317617/2021-4] and International Joint Laboratory "Sentinela" (Fiocruz, UnB, IRD) (grant IRD LMI-Sentinela). The funders had no role in study design, data collection and analysis, decision to publish, or preparation of the manuscript.

**Competing interests:** The authors have declared that no competing interests exist.

and females. Such findings highlight that the strengthening of primary health care combined with reducing socioeconomic, racial, and gender inequalities represents a crucial step to reducing heat-related deaths.

## 1. Introduction

Under continued human-induced climate change, the frequency, duration, intensity, and spatial extent of climate extreme events are continuously rising, including droughts, wildfires, floods, and, in particular, heat waves (HW) [1, 2]. Prolonged periods of excessive heat have become more frequent, intense, and prolonged in many regions across the world and are projected to increase even more considering the future global warming scenarios [3], particularly over tropical regions with low-temperature variabilities, such as Africa, Southeast Asia, and South America (SA) [4, 5]. A wide range of adverse effects of HWs on ecosystems and humans have been identified, including on water consumption [6, 7], increasing wildfires [8, 9], worsening air quality [10], on energy consumption and supply, road and rail transport sectors [11], and human health effects [12–15], with significant social and economic implications [16–18]. Particularly in densely populated cities, human exposure to extreme heat has been exacerbated by the urban heat island (UHI) effect as a consequence of the urbanization process [19]. Consequently, the research on temperature extremes and their adverse impact on human health has expanded in recent years, although the number of studies is not evenly distributed globally. Some regions with more significant vulnerabilities and limited resources to adapt to rising temperatures are underrepresented in the literature, including SA [20]. A recent global assessment on this subject has attributed a significant burden of excessive deaths to heat in several Brazilian cities [21]. However, to our knowledge, the role played by regional, demographic, and social aspects has not yet been sufficiently investigated.

During the last decades, several extreme HWs have been thoroughly investigated, focusing on their main drives and their impacts. The 2003 European summer mega-HW has been deeply studied [22, 23], being characterized by a significant increase in total burnt area [24], high economic losses [25], and substantial impacts on excess mortality, with estimations adding up to 70,000 deaths for the whole summer [26], mainly in France [27]. In the United States, [28] estimated an average of about 5,600 deaths annually attributable to extreme heat stress between 1997 and 2006 and across 297 counties (representing 61.6% of the country's population). In Australia, Sydney alone has an annual average of 117.3 excess deaths (ED) due to excessive heat, strongly related to UHI effects [29]. In Latin America, one of the earliest multi-country studies [30] found a significant association between elevated temperatures and mortality in Santiago, São Paulo, and Mexico City, mostly among older people. Also, in Latin America, [31] investigated the effects on mortality of the HWs that occurred in the center-north region of Argentina during the warm season of 2013–2014. The authors observed increased death risk in 13 of the 18 provinces analyzed. A recent city-level analysis of temperature and mortality among 326 Latin American cities [32] estimated an increase in the risk of death by 5.7% per 1˚C increase in temperature during extreme heat events, mostly among older adults and for cardiovascular and respiratory diseases. In Brazil, around 1800 excess deaths (ED) were related to four HWs events in 2010 and 2012 in Rio de Janeiro, the majority linked to circulatory illnesses [33]. [34] observed a significant association between air temperatures and increased risk of death from cerebrovascular diseases, with different impacts across the country. Moreover, climate model outputs have projected a continuous rise in the

frequency of extreme temperature events in SA [4], increasing population exposure to heat events across the region [35], besides significant amplification effects due to population aging and growth, particularly in densely urbanized areas [36]. In addition, the impacts of HWs on human health tend to be aggravated by compound dry and hot extremes, which have already been observed in heavily populated regions of Southeast Brazil [37, 38].

Despite the increasing number of global studies investigating heat-related mortality risk in Brazilian cities [21], there is still a considerable knowledge gap on the role played by regional, demographic, and socioeconomic aspects [39]. This represents a particular challenge in Brazil, considering the continental size of the country, the variety of different climates to be considered, and an urban population that is rapidly aging. For example, [32] found lower changes in mortality under extreme heat in the tropical areas in northeast Brazil with relatively low-temperature variability compared to temperate regions of southern Brazil. Another recent study by [40] investigated the association between socioeconomic and demographic characteristics and temperature-related human mortality in Latin American cities. According to the authors, income inequality increases mortality from cold events for all ages, in addition to segregation and poverty, which are associated with higher excess mortality due to cold among older people. However, the study reveals the opposite behavior for heat-related mortality, with higher levels of poverty and income inequality associated with lower heat-related mortality. This highlights the need for further investigation, given the increasing importance of urban adaptation to climate change, as discussed by the authors. When addressing climate change in general, populations with less favored socioeconomic conditions regarding access to environmental sanitation, lower educational levels, and income levels tend to suffer more of the adverse effects of these changes and greater exposure to health risks [41, 42]. In this sense, in addition to death due to HW, other negative effects on health can be minimized by adopting measures for the early identification of risk situations. For this, the World Health Organization (WHO) recommends event-based surveillance. This strategy uses the capture and analysis of unofficial information published by social media users as a source of alert [43, 44].

Accordingly, this work presents a comprehensive spatial and long-term temporal scale analysis of the impact of HWs on mortality in Brazil between 2000 and 2018, focusing on the most populous metropolitan regions (MRs) that currently account for 74 million people (circa 35% of the country's current population). It is aimed to evaluate the interplay of intersectionality and vulnerability to heat, considering demographics and socioeconomic heterogeneities, i.e., gender, age, race, and educational level as proxies. The leading causes of heat-related excess death were also investigated, including diseases that are generally not included in epidemiological studies related to HW. Additionally, we performed an event-based surveillance analysis to investigate the early detection and monitoring of mortality outbreaks in Brazil during extreme temperature events. Such tools can be particularly relevant for public health, similar to what has been developed in several countries [45]. Therefore, the findings presented here can help guide public health responses and strategies to adapt to the emerging risks of climate change regionally and nationally.

## 2. Material and methods

### 2.1. Study area

Covering roughly two-thirds of equatorial and subtropical SA, Brazil has a population of 215 million inhabitants, comprising an area of 8,514,877 km$^2$ [46, 47]. Geopolitically, Brazil is divided into 26 states and one federal district grouped into five macroregions (Fig 1): the North, the Northeast, the Central-West, the Southeast, and the South. Equatorial and tropical climates dominate the Brazilian territory in most parts of the North and Central-west, a humid

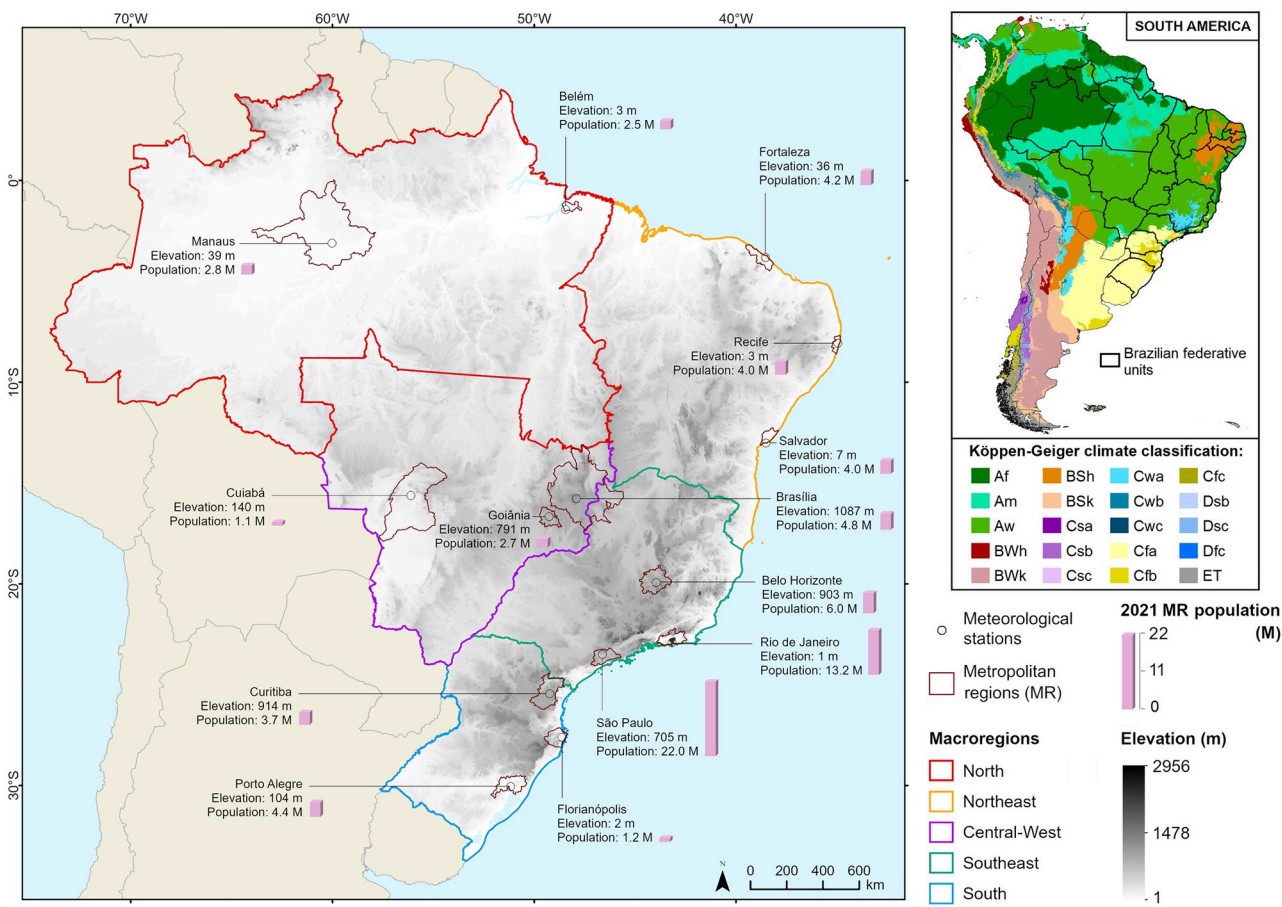

**Fig 1. Map of Brazil divided into five macroregions (Northern, North Eastern, Central-Western, South Eastern, and Southern).** The area and the population of the 14 MRs studied are presented, as well as the location and the elevation of the respective meteorological stations. The Köppen-Geiger climate classification map is also shown in the top-right panel.

subtropical climate in the South, and a semi-arid climate in the Northeast [48]. Nevertheless, the country has experienced several modifications in its climatic limits and variability [49]. Regarding life expectancy, considering the 2000–2018 period, estimations are lower in the Northeast (66.7 and 74.9 years, for males and females, respectively) and North regions (67.3 and 74.0), according to the Brazilian Institute of Geography and Statistics (IBGE), compared to Central-West, Southeast, and South regions (70.1, 71.4 and 72.0 for males, and 76.8, 78.6 and 78.8 for females, respectively).

In the present study, we considered the 14 most populous Brazilian MRs, as defined by the IBGE, namely the MRs of Manaus and Belém, in the North, Fortaleza, Salvador, and Recife, in the Northeast, São Paulo, Rio de Janeiro, and Belo Horizonte, in the Southeast, Curitiba, Florianópolis and Porto Alegre, in the South, and the MRs of Goiânia and Cuiabá, and the Federal District and its Economic Development Integrated Region (hereafter MR of Brasília), in the Central-West (Fig 1). These MRs comprise 74 million people, representing circa 35% of Brazil's 2021 population, most of them in the MRs of Rio de Janeiro (13.0 million people) and São Paulo (20.4 million people) (Table 1).

**Table 1. List of MRs analyzed in this study, sorted by the total population in 2021, including their respective macroregion, geographic coordinates, and altitude.** Population data was provided by the Brazilian Institute of Geography and Statistics—IBGE.

| Metropolitan Region | 2021 Population (million people) | Meteorological station location | | Macroregion | Altitude (m) |
|---|---|---|---|---|---|
| | | Lat S (°) | Lon W (°) | | |
| São Paulo | 22.0 | 23.50 | 46.62 | Southeast | 705 |
| Rio de Janeiro | 13.2 | 22.91 | 43.17 | Southeast | 1 |
| Belo Horizonte | 6.0 | 19.93 | 43.95 | Southeast | 903 |
| Brasília | 4.8 | 15.78 | 47.92 | Central-West | 1,087 |
| Porto Alegre | 4.4 | 30.05 | 51.17 | South | 104 |
| Fortaleza | 4.2 | 15.78 | 47.92 | Northeast | 36 |
| Recife | 4.0 | 8.06 | 34.96 | Northeast | 3 |
| Salvador | 4.0 | 13.01 | 38.51 | Northeast | 7 |
| Curitiba | 3.7 | 25.45 | 49.23 | South | 914 |
| Manaus | 2.8 | 3.10 | 60.02 | North | 39 |
| Goiânia | 2.7 | 16.67 | 49.26 | Central-West | 791 |
| Belém | 2.5 | 1.44 | 48.44 | North | 3 |
| Florianópolis | 1.2 | 27.60 | 48.62 | South | 2 |
| Cuiabá | 1.1 | 15.62 | 56.11 | Central-West | 140 |

## 2.2. Data

Daily maximum and minimum surface air temperature data (1970–2020) from meteorological stations were provided by the Brazilian National Institute of Meteorology (INMET, https://portal.inmet.gov.br/, accessed 04 October 2021) and the Brazilian Institute of Air Space Control (ICEA, https://www.icea.decea.mil.br/, accessed 06 April 2022), according to the data availability (Table 1 and Fig 1). All meteorological stations are located inside the urban core of each MR. For temperature data, the percentage of missing values along the time series was below 10%, varying from 11% to 17% only in three (Salvador, Florianópolis, and Fortaleza). It is worth noting that the WMO (World Meteorological Organization) recommends a threshold of 20% for missing data, below which they are considered acceptable [50]. No significant differences were observed in missingness between cold and warm months, and missing data are spread throughout the entire historical series.

Daily mortality data from the Brazilian Health Informatics Department (DATASUS) for the 2000–2018 period were provided by the Data Science Platform applied to Health (PCDaS) of the Oswaldo Cruz Foundation (https://pcdas.icict.fiocruz.br/, accessed 04 October 2021). All-cause deaths were considered, excluding those related to external causes such as accidents or murders, and all municipalities within each MR were included, adding up to 9,357,684 deaths analyzed. Regarding death records, the percentage of missing data was low for age (ranging from 0.1% in Florianópolis to 2.0% in Fortaleza) and sex information (ranging from 0.0% in São Paulo to 0.2% in Salvador). However, race and education presented a higher percentage of missing data, mainly due to the higher number of records filled as "ignored" in these categories, compared to gender and age. For race, missingness was below 10% for most MRs (1.73% to 8.82%), except Fortaleza (13.8%). For education, the percentage of missing data was more significant (between 10% and 30% for most of the MRs), being higher for Recife (35.2%), Porto Alegre (42.1%), and Goiânia (52.1%), which reflects the significant uncertainties in the results for these MRs. Ill-defined deaths and unknown causes of mortality (ICD XVIII) ranged from 1.3% to 9.3% for all MRs, except for Manaus, which reached 16.60% of data. The high proportional mortality from unspecified and ill-defined causes of death in the

State of Amazonas (where the MR of Manaus is located) has already been reported by a previous study [51], showing to be associated with space-time dimensions demographic factors and socioeconomic status and medical assistance at the time of death. However, the authors also showed a significant reduction in the fraction of ill-defined causes in the state between 2006 and 2012.

Mortality data were disaggregated by gender and age group—namely children (< 10 years old), young (10–44), adults (45–64), older (65–74), and eldest (> 74 years old), following previous studies in Brazil [52], cause of death (according to the 10th revision of the International Statistical Classification of Diseases and Related Health Problems, ICD-10), besides race and educational level. DATASUS provides all this information in the individual death records.

Educational level was divided into low (less than four years) and high (12 or more years). Although this division results in a reduction in the number of death records (since it removes individuals who fell between the low and high education categories), after a sensitivity test, we observed that this makes the results more robust, as it allows us to compare extreme categories with each other.

Regarding racial classification, three groups were considered in the analysis: whites, blacks, and browns ("pardos"). Despite the racial mix in Brazil, several studies on different fields of public health consider the racial split of the population [53]. Most of them have shown that race is a strong predictor of variability in mortality [54], irrespective of socioeconomic status [55]. The ternary system composed of blacks, browns, and whites has been used in previous studies on racial inequalities in health in Brazil [56], unlike the binary Black-White dichotomy most used in countries like the USA. However, according to article IV of Federal Law nº 12.288 of 2010, which creates the Statute of Racial Equality, "Negros" are: "the group of people who declare themselves black or brown, according to the color or race used by the IBGE, or that adopt analogous self-definition". In Brazil, particularly in urban areas, "Negros" are the racial group of the population that more frequently live under conditions of deprivation of services, such as sanitation, higher rates of violence, unemployment, inadequate housing, etc., being disproportionately affected by social and environmental injustices [57]. Accordingly, we combine the black and brown groups into one. Indigenous and yellow groups were not included in the analysis since they represent less than 2% of Brazil's population, making robust statistical analysis unfeasible. The IBGE provided all the population estimates (https://www.ibge.gov.br/, accessed 15 February 2022).

## 2.3. Excess heat factor and characterization of HWs

The Excess Heat Factor (EHF), described in detail by [58], was used as an index to identify and classify HWs according to frequency, duration, and intensity. The EHF has been widely used [33, 38, 59, 60] and has been shown to have a significant advantage in predicting the impacts of HW on human health, constituting a good health-relevant heat exposure indicator [61].

The EHF is calculated based on two indices. The first is the significance index ($EHI_{sig}$), computed as the difference between three-day-averaged daily mean temperature (DMT) and the 95th percentile of this DMT calculated across 1981–2010, using all days of the year in the calculation (hereafter $T_{95}$). The second is the acclimatization index ($EHI_{accl}$), which considers the human physical ability to adapt to temperature anomalies and is calculated as the difference between the three-day-averaged DMT and the average DMT over the previous 30 days. So, these two heat indices are computed as follows:

$$EHI_{sig} = \left(T_i + T_{i+1} + T_{i+2}\right)/3 - T_{95} \tag{1}$$

and:

$$EHI_{accl} = \left(T_i + T_{i+1} + T_{i+2}\right)/3 - (T_{i-1} + T_{i-2} + \ldots + T_{i-30})/30 \qquad (2)$$

Where $i$ corresponds to the first day under HW. Finally, the EHF is calculated as the product of the two EHI:

$$EHF = EHI_{sig} \times max(1, EHI_{accl}) \qquad (3)$$

Thus, the EHF has the same sign as the significance index and units of $[^\circ C^2]$. If the $EHI_{sig}$ is positive (and therefore, the EHF is also), then the three days are considered HW days. Instead of only the daily maximum temperature, the EHF index incorporates the minimum temperature by calculating DMT. According to [58], thermal accumulation depends on the extent to which heat is dissipated during the night. Accumulation of undissipated heat results in overheating. After a day of high temperature, this excess determines the load that impacts vulnerable people.

The $EHI_{accl}$ has the property of amplifying the EHF. The classification of HW intensity is obtained by considering the 85[th] percentile of all the positive EHF values ($EHF_{85}$) as a threshold since the generalized Pareto distribution describes it well. Therefore, days with negative EHF (i.e., no HWs) are not considered for calculating $EHF_{85}$. HWs with EHF<$EHF_{85}$ are designated low-intensity, HWs with EHF>$EHF_{85}$ as severe and HWs with EHF>3×$EHF_{85}$ as extreme HWs. Although we used the method described by [58] to classify HWs, in some parts of this paper, severe and extreme events will be interpreted as a single category, given the low number of extreme HWs and similar results regarding impacts on mortality.

From the HWs calculated using the EHF index, we determined the total annual number of events for each MR. Then, we performed a trend analysis using the Ordinary least squares Linear Regression method with the scikit-learn tool, a free software machine learning library for the Python programming language.

## 2.4. Excess mortality estimates

HW-related excess mortality was estimated based on the observed-to-expected deaths (O/E) ratio, a simple method largely used to estimate the burden of heat-related mortality [33, 38, 62]. The O/E ratio has the advantage of allowing to compare directly the number of deaths observed during short periods, e.g. during a HW, with the average number of deaths in similar periods (with the same duration and at the same time of the year), so values higher than 1 means excess mortality, as follow:

$$\left(\frac{O}{E}\right)_{ij} = \frac{M_{ij}}{\left(M_{i1} + M_{i2} + \ldots + M_{i,j-1} + M_{i,j+1} \ldots + M_{i,k-1} + M_{i,k}\right)/(k-1)} \qquad (4)$$

Considering the $i^{th}$ HW that occurred in the $j^{th}$ year, $M_{ij}$ is the observed number of deaths over all HW days. Expected mortality is calculated considering the average number of deaths over periods of the same duration using the same Julian days of the year in previous ($M_{i1}$, $M_{i2}$,..., $M_{i,j-1}$) and subsequent ($M_{i,j+1}$, ..., $M_{i,k-1}$, $M_{i,k}$) years, ranging from 1 (in this case representing the year 2000) to $k$ (2018). Reference periods under HW condition (i.e., days that fell under an HW), either in an earlier or later year, were not included in the calculation of expected mortality. Moreover, the number of deaths was normalized by annual population to minimize population growth effects. Therefore, the O/E ratio and the corresponding confidence interval (C.I. 95%), as proposed by [63], were calculated, and for events in which it was significantly higher than unity, the excess mortality was estimated. Data processing and

quantitative analysis were performed using the *Pandas* open-source data analysis and manipulation tool, built on top of the *Python* programming language. Data visualization was performed using *Seaborn* and *Matplotlib Python* data visualization libraries.

The O/E ratio and excess mortality were estimated for total all-cause natural deaths, stratified by age, sex, race, and education, and also for specific ICDs. We also conducted analyses of sex differences in heat-related mortality stratified by age. Despite the results being statistically significant for the MRs with large populations, dividing death records into several subgroups reduced the dataset for most MRs, making the analysis unfeasible. Moreover, the results comparing analysis segregated by sex and age with non-segregated analyses for MRs with large populations were comparable within the confidence interval. Therefore, we chose not to include this analysis stratified by sex and age jointly. To avoid any bias related to missingness, for both race and education data, we normalize the daily number of deaths in different categories (black/brown/white or low/high educational level), dividing by the number of records filled with respectively race/education information, and then multiplying by the total number of deaths. Finally, based on the values of the O/E ratio, we identified the HWs with the large mortality increase for each MR these events according to the excess deaths, duration and intensity, and the interval after the previous HW for further analysis.

## 2.5. Event-based surveillance as a risk identification strategy: Epidemic Intelligence from Open Sources (EIOS)

An event-based surveillance analysis using rumor search was performed using Epidemic Intelligence from Open Source (EIOS), which is a tool implemented by the World Health Organization (WHO) to identify HW rumors on social media [43, 44], to identify if the HWs registered in Brazil were mentioned in some rumor on media. A rumor is any information circulating, usually from unofficial sources, about reports of disease outbreaks, adverse events, impacts of health products or services, or other information about a negative effect on health [44, 64–67]. EIOS, through internet-based media monitoring, gathers epidemic intelligence information from different sources, including traditional online media and specific social media sources, government and official websites, news aggregators, blogs and expert groups, and collaboration, among others. The system automatically collects, filters, and processes news from relevant online sources, categorizing them based on key terms, languages, regions, periods, etc. Electronic tools have been increasingly used as monitoring and early warning mechanisms in the risk management strategy of public health emergencies and surveillance processes [43, 68, 69]. The period covered by the search was 2000–2018, using the terms "heat waves" + "Brazil" + "health", all news categories, all languages, and placing only Brazil in the filter for the country cited.

## 3. Results

### 3.1. Long-term trends in frequency, duration, and intensity of HWs in Brazil

Overall, the number of HWs from 1970 to 2020 has increased in all Brazilian MRs, particularly for low-latitude regions (Fig 2, Table 2). In northern and northeastern MRs, the median annual HW frequency throughout the first two decades (the 1970s and 1980s) was lower than one HW year$^{-1}$. Contrastingly, over the last two decades (the 2000s and 2010s), there has been a significant increase, ranging from 3.5 HWs year$^{-1}$ in the MR of Salvador to 11 HWs year$^{-1}$ in Belém. Although less intense, similar behaviors were observed in the south, southeast, and central-western MRs. Statistically significant (C.I. 95%) increasing trends for HW frequency were

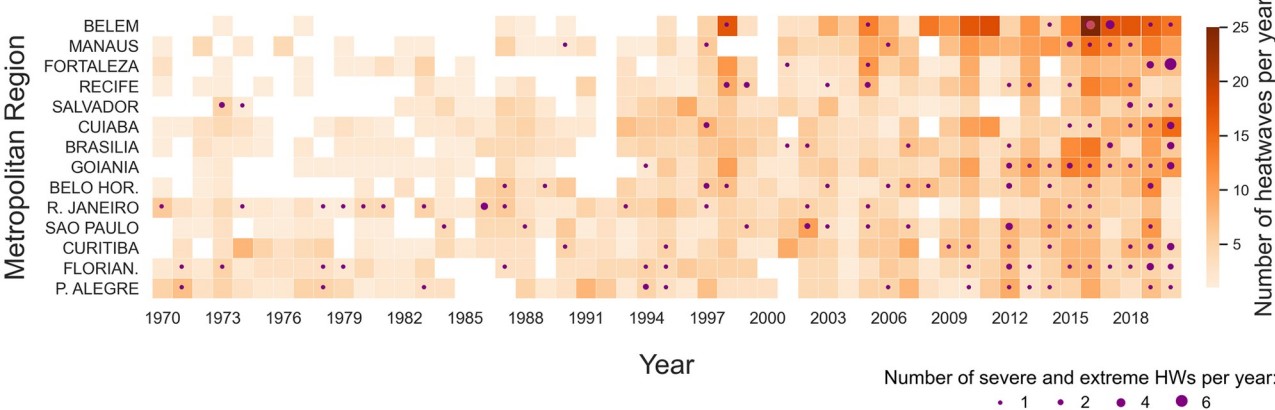

**Fig 2. Temporal evolution of the annual number of HWs during the period 1970–2020 at the Brazilian metropolitan regions (MR) sorted by latitude (decreasing with the distance from the Equator).** The color bar represents the total number of HWs per year and the size of the purple circles represents the number of severe and extreme HWs.

observed in most MRs (except for MRs of Rio de Janeiro and Porto Alegre) (Table 2), ranging from 0.06 HW year$^{-1}$ (Florianópolis) to 0.36 HW year$^{-1}$ (Belém), with high values also in Manaus (0.23 HW year$^{-1}$) and Goiânia (0.20 HW year$^{-1}$). Likewise, events classified as severe and extreme, which used to be sporadic, have become more frequent. In the last five years (2016–2020) of the historical series (Fig 2), all MRs registered at least one severe/extreme HW, whereas, throughout the 1970s and early '80s, its frequency and spatial coverage were significantly lower, having occurred only in the MRs of Salvador, Rio de Janeiro, Florianópolis, and Porto Alegre. Moreover, HW duration has also increased. Throughout the 1970s and '80s, the median HW duration varied between 3 and 5 days (Table 2), while in the 2000s and '10s, the range of variability rose considerably, ranging between 4 and 6 days. It is worth noticing that

**Table 2. Statistical description of frequency and duration of HWs over 1970–1990 period, and also later during 2000–2020 period for the Brazilian MRs, including median and interquartile intervals.** Trends in the annual HWs number are also presented. Underlined values indicate a non-statistically significant (C.I. 95%) trend based on the Mann-Kendall Trend Test. The MRs are sorted by latitude (decreasing with the distance from the Equator).

| Metropolitan Region | Frequency of HW [events year$^{-1}$] | | | Duration of HW [days] | |
|---|---|---|---|---|---|
| | 1970–1990 (median and IQ) | 2000–2020 (median and IQ) | trend (slope) | 1970–1990 (median and IQ) | 2000–2020 (median and IQ) |
| Belém | 0.0 (0.0–0.0) | 11.0 (5.0–16.2) | 0.36 (0.04) | 4.0 (3.0–5.0) | 5.0 (4.0–8.2) |
| Manaus | 0.5 (0.0–1.0) | 9.0 (5.8–10.2) | 0.23 (0.03) | 4.0 (3.0–5.0) | 6.0 (4.0–10.0) |
| Fortaleza | 0.0 (0.0–1.0) | 4.0 (2.0–7.2) | 0.13 (0.02) | 5.0 (3.0–6.0) | 6.0 (3.0–9.0) |
| Recife | 0.5 (0.0–1.0) | 3.5 (2.0–8.0) | 0.15 (0.02) | 4.0 (3.0–5.8) | 4.0 (3.0–6.0) |
| Salvador | 0.5 (0.0–2.0) | 3.5 (2.0–5.2) | 0.09 (0.02) | 5.0 (4.0–7.0) | 5.0 (3.0–8.0) |
| Cuiabá | 2.0 (1.0–3.0) | 5.0 (3.0–6.5) | 0.14 (0.02) | 4.0 (3.0–5.0) | 4.0 (3.0–6.0) |
| Brasília | 2.0 (1.0–2.2) | 4.0 (3.0–7.0) | 0.14 (0.02) | 4.0 (3.0–5.0) | 6.0 (4.0–10.0) |
| Goiânia | 1.0 (0.0–1.0) | 6.5 (3.8–8.2) | 0.20 (0.02) | 5.0 (3.0–7.2) | 6.0 (4.0–10.0) |
| Belo H. | 0.5 (0.0–2.2) | 5.0 (4.0–7.0) | 0.13 (0.02) | 5.0 (3.0–6.0) | 5.0 (4.0–7.0) |
| R. Janeiro | 2.5 (2.0–4.0) | 4.0 (2.8–5.0) | <u>0.03 (0.02)</u> | 4.0 (3.0–5.0) | 5.0 (4.0–7.0) |
| São Paulo | 1.0 (0.0–2.2) | 5.5 (4.0–7.0) | 0.12 (0.02) | 4.5 (3.0–5.8) | 4.0 (3.0–7.0) |
| Curitiba | 3.0 (1.0–3.2) | 6.0 (4.0–8.0) | 0.09 (0.02) | 4.0 (3.0–6.0) | 5.0 (3.0–6.0) |
| Florianóp. | 2.5 (2.0–3.0) | 4.0 (3.0–5.0) | 0.06 (0.02) | 5.0 (3.2–6.0) | 5.0 (4.0–8.0) |
| Porto Alegre | 3.0 (2.0–5.0) | 4.5 (3.0–5.2) | <u>0.04 (0.02)</u> | 5.0 (4.0–7.0) | 5.0 (4.0–7.8) |

despite the small increase in the median values, the interquartile range of HW duration was much larger in the last two decades, particularly in Manaus, Brasília, and Goiânia (4–10 days), when compared to the first two decades. Regarding seasonal distribution, these events showed different patterns throughout the year according to the regions (Figure in S1 Fig). For the northeast, southeast, and southern MRs, the HW frequency was larger in the austral summer wet season (December to February), with slightly different behavior for severe and extreme HWs in the MRs of São Paulo and Belo Horizonte, more frequent in September and October. For the north and central-western MRs, the HWs are mostly concentrated in September and October, although the seasonal profile in Belém is more homogeneous throughout the year.

## 3.2. HW-related ED in Brazilian urban areas

Considering the nearly two-decade period spanning between 2000 and 2018, the O/E ratio during HWs was calculated (Fig 3). In the MRs of Florianópolis, Salvador, and Cuiabá, only a small fraction of these events (6–12%) resulted in a significant increase in mortality (red circles in Fig 3), while in the MRs of Rio de Janeiro, São Paulo, Porto Alegre, and Belo Horizonte it corresponds to 41–68% of the total of HWs identified between 2000 and 2018.

The cumulative HW-related ED between 2000 and 2018 (bars at the bottom of Fig 3) was estimated as 48,075 (40,448–55,279) ED (Table 3), considering the 14 MRs. The most populous MRs (São Paulo and Rio de Janeiro) also had the highest absolute amounts of ED (14,850 and 9,641, respectively). However, a different rank emerges if excess mortality is normalized by annual population and number of days under the HW regime. Considering the 2000–2018 period, the MRs of Rio de Janeiro, Porto Alegre, Belém, Cuiabá, and Recife presented the highest normalized heat-related mortality rates, adding up to 1.30–2.06 ED per 1 million inhabitants per HW-day. Contrastingly, if only the last five years are considered (2014–2018), the MRs of Recife, Belém, Porto Alegre, Cuiabá, and São Paulo appear as the most impacted, adding up to 2.34–3.01 ED per 1 million inhabitants per HW-day. The comparison between normalized mortality rates throughout the 2000–2018 and the 2014–2018 period showed an

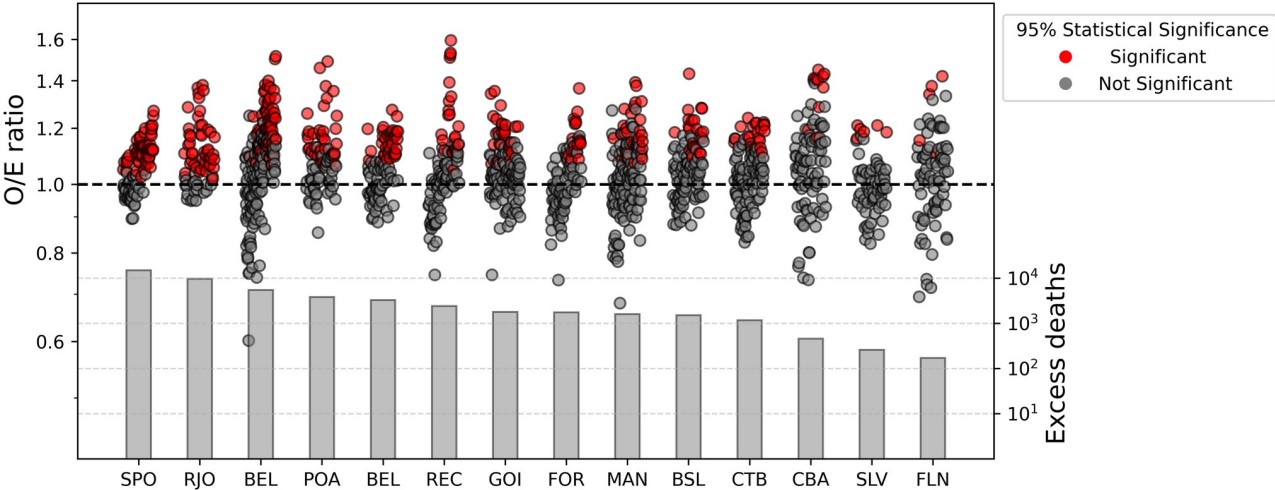

**Fig 3. Heat-related excess deaths in Brazilian metropolitan regions.** Observed to expected (O/E) ratio (dots, left y-axis) and estimation of excess mortality (bars, right y-axis in logarithmic scale) during HWs over the 2000 to 2018 period for the MRs of Belém (BEL), Manaus (MAN), Fortaleza (FOR), Salvador (SLV), Recife (REC), Brasília (BSL), Cuiabá (CBA), Goiânia (GOI), Belo Horizonte (BHE), São Paulo (SPO), Rio de Janeiro (RJO), Porto Alegre (POA), Curitiba (CTB), and Florianópolis (FLN).

**Table 3. HW-related excess deaths (excluding external causes) over the 2000–2018 period and regarding the 14 most populated metropolitan regions in Brazil sorted by ED.** The second column highlights the total number of excess natural deaths estimated for the whole period. The third column shows the average mortality rate normalized by 1,000,000 people and by the number of days under HW conditions.

| Metropolitan Region | Total excess deaths (2000 to 2018) | Average per HW-day (per 1,000,000 people) | |
|---|---|---|---|
| | | **2000–2018** | **2014–2018** |
| São Paulo | 14,850 (13,814–15,013) | 1.28 | 2.18 |
| Rio de Janeiro | 9,641 (8,845–9,800) | 2.06 | 1.98 |
| Belém | 5,429 (4,475–6,517) | 1.46 | 2.44 |
| Porto Alegre | 3,810 (3,025–4,323) | 1.97 | 2.34 |
| Belo Horizonte | 3,263 (2,526–3,624) | 1.06 | 1.89 |
| Recife | 2,419 (1,885–2,955) | 1.30 | 3.01 |
| Goiânia | 1,776 (1,265–2,555) | 0.86 | 0.81 |
| Fortaleza | 1,735 (1,271–2,226) | 0.63 | 1.65 |
| Manaus | 1,585 (899–2,294) | 0.53 | 0.83 |
| Brasília | 1,513 (1,160–2,137) | 0.45 | 0.75 |
| Curitiba | 1,165 (637–1,800) | 0.55 | 1.02 |
| Cuiabá | 458 (373–872) | 1.32 | 2.20 |
| Salvador | 258 (170–572) | 0.17 | 0.22 |
| Florianópolis | 171 (104–591) | 0.29 | 0.20 |
| **Total** | **48,075 (40,448–55,279)** | | |

increase in most MRs, except for the MRs of Rio de Janeiro, Goiânia, Salvador, and Florianópolis, which seems to be stable.

## 3.3. Age, gender, racial, and socioeconomic disparities in the vulnerability to HWs

The heat-related mortality was further investigated considering the split by age, gender, and socioeconomic subgroups, namely by race and educational level. For this purpose, the MRs of Salvador and Florianópolis were not included due to their small number of absolute ED throughout the 2000–2018 period (<300 ED), undermining the statistical significance of the results when the split among subgroups is considered. Since the population in each category is different, the excess mortality estimations are presented in percentage and observed-to-expected ratio values.

As expected, the fraction of ED among age groups (Fig 4a) was dominated by older people, although some regional differences can be observed. In the south, southeast, and central-western MRs, the percentage attributed to under-65 subgroups represented only 6–19% of ED, while in the north and northeastern MRs, it reached larger values (17–25%). The large fraction of excess heat-related deaths among older people over 65 years (75–94% of excess deaths) contrasts with the distribution of total mortality from natural causes (S2 Fig). When considering all natural deaths from 2001 to 2018, the older and eldest groups represent less than 60% of the observed mortality, ranging from 47.65% in the MR of Manaus to 59.90% in the MR of Rio de Janeiro. Females appear as the most affected group, with O/E values consistently higher (1.15–1.36) than those observed for males (1.07–1.23) (Fig 4b). Interestingly, such gender inequalities were more expressive in the MRs of Porto Alegre, Belo Horizonte, São Paulo, Rio de Janeiro, Recife, and Goiânia.

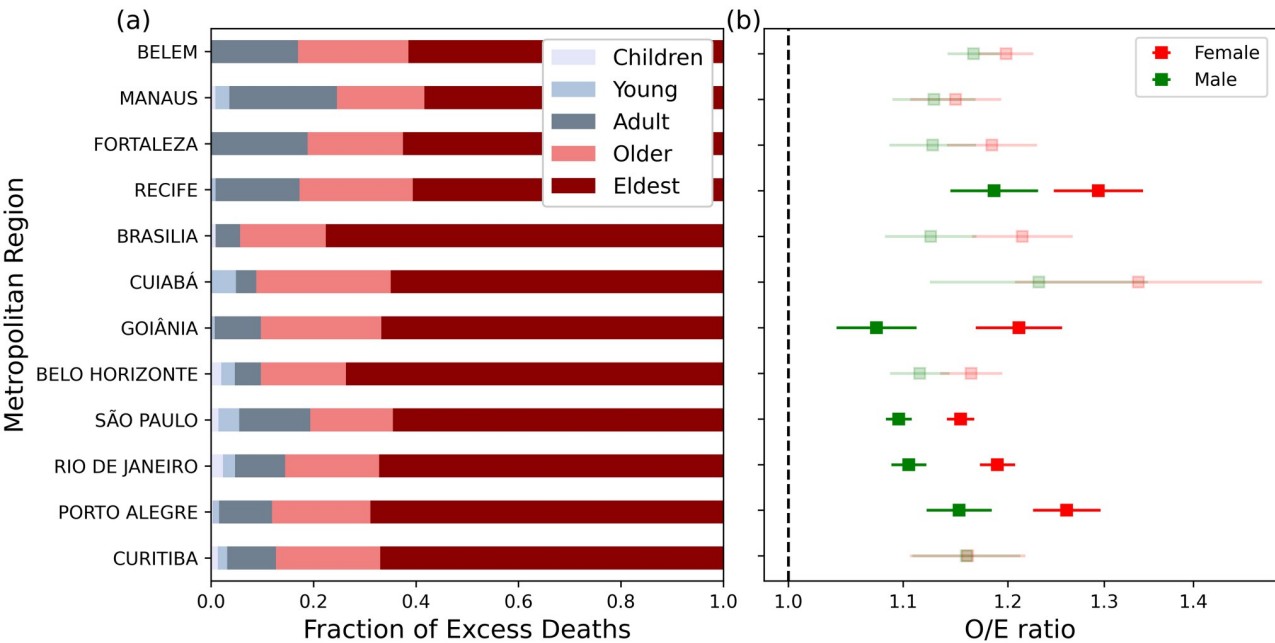

**Fig 4. Age and gender differences in heat-related mortality.** (a) Fraction of excess mortality according to age, including children (<10), young (10–44), adult (45–64), older (65–74), and eldest (>74 years old); and (b) O/E mortality ratio for males (green) and females (red) in the Brazilian metropolitan regions. The error bar represents the confidence intervals (C.I. 95%) of the O/E values.

In terms of the socioeconomic level (Fig 5a and 5b), the O/E ratio among 65 years old and older people was significantly higher for the low educational level (1.21–1.86) than for the high educational level sub-group (0.95–1.29), for both genders in the MRs of Belém, Recife, São Paulo, and Rio de Janeiro. Additionally, larger O/E values are observed in the less educated people for females in the MR of Manaus (1.54, compared to 0.94 for high educational level) and males in Porto Alegre (1.47, compared to 1.04 for high educational level). In the MRs of Fortaleza, Brasília, Cuiabá, Goiânia, Belo Horizonte, and Curitiba, the O/E ratio for these sub-groups was not statistically distinguishable at a 95% confidence level.

Considering the split in terms of race (Fig 5c and 5d), black/brown people (among those 65 years old and older) showed O/E ratios statistically higher (1.33–2.30) than the observed for white people (1.16–1.44) in the MRs of Belém, Recife, Brasília, Goiânia, São Paulo, and Rio de Janeiro for both genders. In the MRs of Manaus, Belo Horizonte, and Curitiba, such racial disparity was observed for males (with O/E between 1.24 and 1.31 for white people and O/E between 1.46 and 2.37 for black/brown people). In the MR of Cuiabá, the results were statistically significant for females, with the O/E ratio increasing from 1.60 among whites to 3.25 for black and brown people. There were no significant differences in the MRs of Fortaleza and Porto Alegre, neither for men nor for women. However, in none of the MRs, the increase in mortality of whites exceeds that of blacks and browns.

### 3.4. The leading causes of HW-related deaths across the Brazilian MRs

In this section, we investigated the leading causes of excess mortality during the excessive mortality HWs between 2000 and 2018 (Fig 6). Diseases caused by a failure of the circulatory and respiratory systems and related to neoplasms dominated ED, accounting for 46–70% of total ED (Fig 6). Regarding respiratory diseases, the MRs of Goiânia, Cuiabá, São Paulo, Fortaleza,

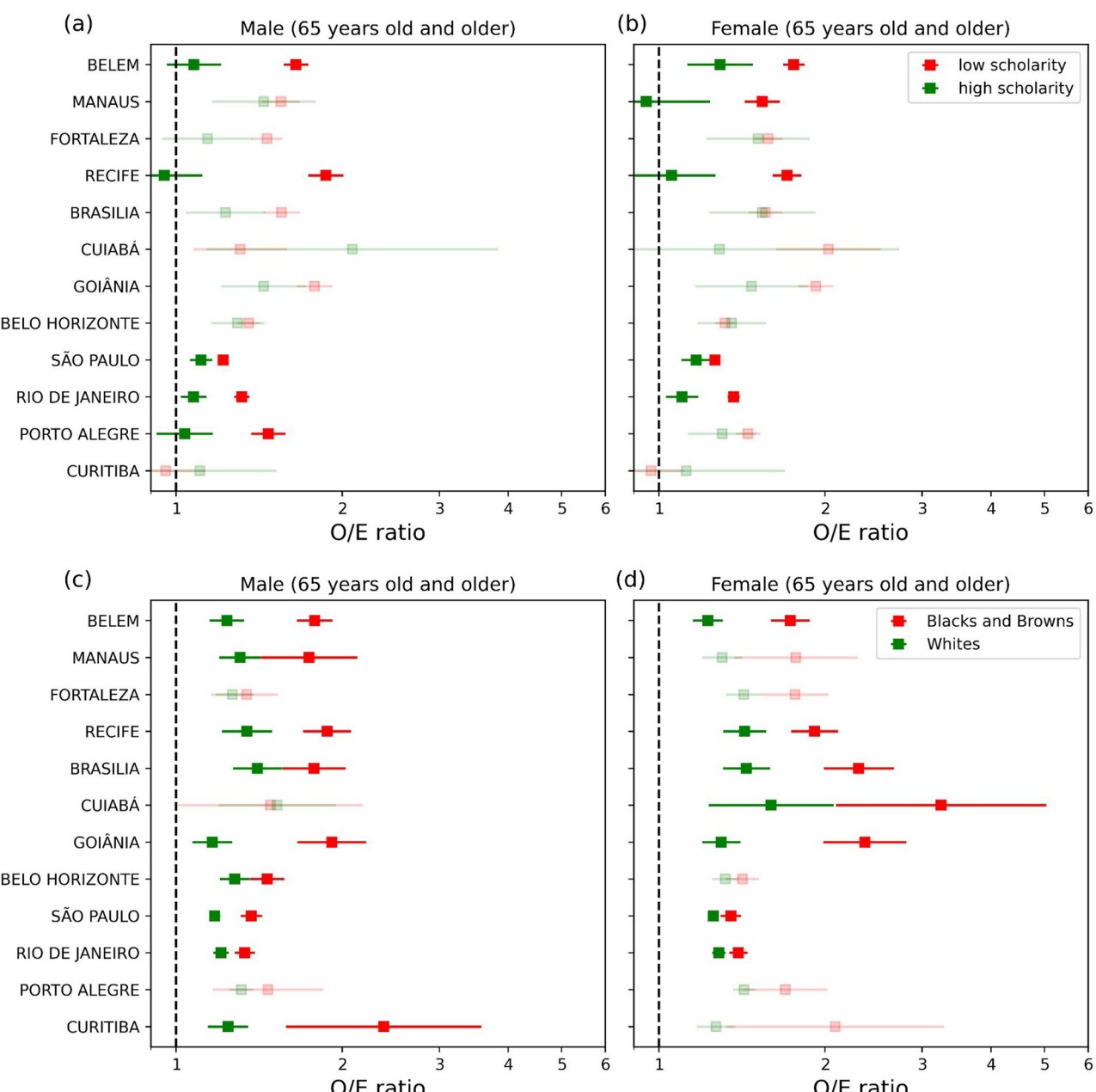

**Fig 5. Socioeconomic differences in heat-related mortality.** O/E mortality ratio for males and females >65 years old at (a and b) different educational levels (high/low) and (c and d) different racial subgroups (whites/blacks and browns). The error bar represents the confidence intervals (C.I. 95%) of the O/E values. Low opacity indicates that differences between O/E for subgroups were not statistically significant.

and Rio de Janeiro had the highest fraction of mortality attributed to this cause of death (20–25% of ED). Circulatory diseases had the highest contributions in the MRs of Rio de Janeiro (31%) and Cuiabá (32%). Neoplasms reach the highest percentage of ED in the MRs of Brasília, Belo Horizonte, Goiânia, and Manaus (28–30%). To a lesser extent, nervous system diseases accounted for just over 10% of heat-related excess mortality in the MRs of Curitiba, Brasília, Porto Alegre, and Belo Horizonte. Genitourinary system-related deaths accounted for ~10% of ED in Belo Horizonte, São Paulo, Rio de Janeiro, and Recife.

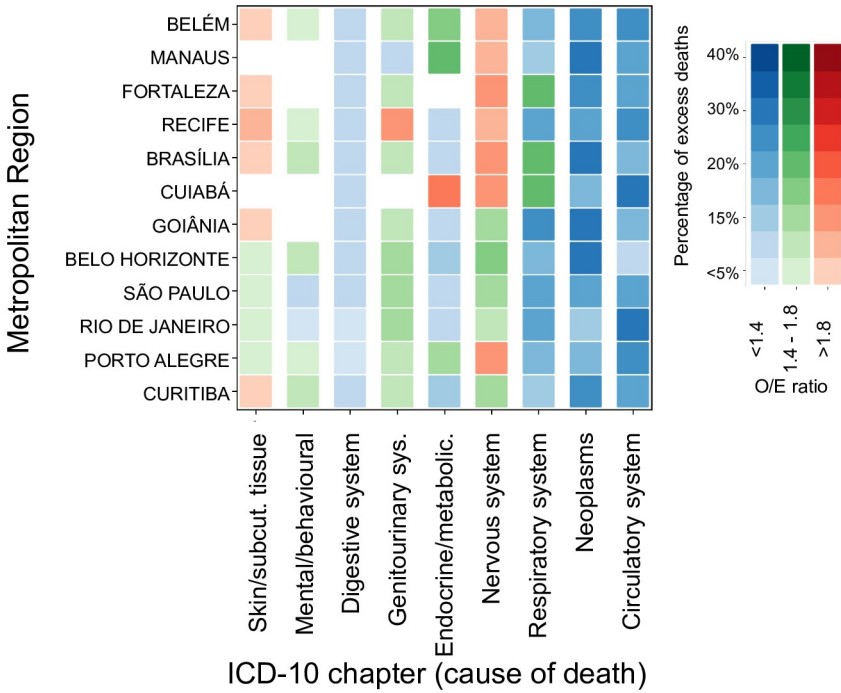

**Fig 6. The leading causes of excessive mortality during HWs across the largest MRs in Brazil.** The colors represented the range of O/E values, including blues (1 <O/E ≤1.4), greens (1.4< O/E ≤1.8), and reds (O/E> 1.8). The percentage of HW-related ED attributed to each cause of death is represented by the top color bars. Blank/not shown values are not statistically distinguishable (C.I. 95%).

When considering the relative mortality increase, skin and subcutaneous tissue diseases are highlighted, with the highest O/E ratio in almost all MRs. High O/E values were also observed for nervous system diseases, ranging from 1.5 to 2.5 across almost all the MRs, and diseases of the genitourinary system, which presented a significant increase in the O/E mortality ratio (1.3–1.9) in all regions studied, except for MR of Cuiabá. Despite their overall small increase, diseases of the digestive system also presented a statistically significant increase in mortality rates in all MRs. In the case of endocrine, nutritional, and metabolic diseases, O/E values were higher than one on most MRs and notoriously high in Belém, Brasília, Recife, and Cuiabá (1.9–2.5). For mental and behavioral disorders, O/E values at 1.2–1.3 were observed in Rio de Janeiro and São Paulo, at 1.5–1.6 in Porto Alegre, Recife, Curitiba, Brasília, and Belo Horizonte, and reaching 1.8 in the MR of Belém.

## 3.5. HW with the highest impact on mortality by region

Since the HW-related ED were not homogeneous, varying widely across the events assessed over the 2000–2018 period (Fig 3), we further investigated the HWs with the largest O/E values for each MR (Table 4). With a few exceptions, most of them were relatively short-term events, with just a few days of duration (<6 days), and classified as low-intensity. The relative increase in mortality during these events varied between 21% and 60% of excess deaths. The MRs of Belém and Recife reached the largest percentage of ED (51 and 60%) in 2018 and 2016 HWs, respectively. Among the 14 MRs, 11 of them had the events with the highest percentage of increase in the number of deaths occurring in the last five years analyzed (2014–2018). The events are concentrated in the dry months for some of the MRs in the North and Central-West

**Table 4. Description of the HWs with the greatest increase in excess mortality in each MR, including date of occurrence (year/month/day), observed increase in mortality, duration and intensity of the HW, and time interval after previous HW at the same location.** The HWs are sorted by excess mortality in percentage.

| Metropolitan Region | HW start date (y/m/d) | Excess deaths (%) | Excess deaths (deaths day$^{-1}$) | HW duration (days) | HW intensity | Interval after previous HW (days) |
|---|---|---|---|---|---|---|
| Recife | 2016/03/05 | 60 (45–75) | 41 (31–53) | 10 | Low | 8 |
| Belém | 2018/10/14 | 51 (25–84) | 17 (8–28) | 5 | Low | 13 |
| Porto Alegre | 2014/01/17 | 49 (40–58) | 32 (27–38) | 28 | Severe | 11 |
| Cuiabá | 2015/09/30 | 45 (2–106) | 6 (1–14) | 4 | Low | 15 |
| Brasília | 2012/10/02 | 43 (20–71) | 14 (7–24) | 6 | Low | 22 |
| Florianópolis | 2016/01/29 | 42 (1–101) | 6 (1–14) | 4 | Low | 5 |
| Manaus | 2016/04/15 | 39 (6–83) | 11 (2–24) | 3 | Low | 98 |
| Rio de Janeiro | 2014/01/03 | 38 (25–52) | 92 (62–126) | 3 | Low | 5 |
| Fortaleza | 2018/01/15 | 37 (11–68) | 19 (6–35) | 3 | Low | 19 |
| Goiânia | 2005/04/06 | 35 (5–75) | 12 (2–25) | 3 | Low | 174 |
| Belo Horizonte | 2016/02/01 | 27 (10–48) | 21 (7–36) | 4 | Low | 92 |
| São Paulo | 2018/12/11 | 27 (22–32) | 81 (65–97) | 13 | Low | 29 |
| Curitiba | 2009/11/18 | 24 (1–48) | 11 (1–23) | 3 | Low | 17 |
| Salvador | 2015/11/24 | 21 (1–45) | 11 (1–19) | 3 | Low | 233 |

(Belém, Cuiabá, and Brasília) and in the summer wet months for most of the South, Southeast, and Northeastern MRs. Moreover, most of these events occur in short time intervals after the previous recorded HW (<30 days), which is more evident among those with the highest O/E.

### 3.6. Health surveillance during HWs in Brazil

The event-based surveillance analysis with EIOS resulted in only 42 occurrences that addressed HW as a climate effect in the period studied (2000–2018). Signals captured with EIOS pointed out that information about HWs mentioning Brazil was only recorded from 2014 onwards, mostly from 2016 onwards. The year 2018 concentrates 58.3% of publications in the period, followed by 2016 (19.4%) and 2017 (13.9%). Most reported rumors were published in Brazil (59.7%), followed by Canada (19.4%), and sources from Germany, England, Portugal, the USA, Mexico, Angola, Ecuador, and Venezuela published 20.8%. Despite the increasing number of rumors detected with EIOS, none of them matched any of the high-mortality HWs presented in Table 4, even though they were the events in which the largest increase in mortality (highest O/E ratio) was observed in each MR.

## 4. Discussion

### 4.1. Heat-exposure of urban population

This work presents a comprehensive analysis of heat-related mortality in the most populous Brazilian urban areas, with extensive geographical coverage, providing novel knowledge in the climate change and health interface. Following previous studies in the country, we observed an increase in the frequency of HWs affecting the Brazilian MRs across the 1970–2020 period [70]. Increasing trends in the annual number of HWs were observed in all MRs, although not statistically significant in Rio de Janeiro, similar to the observed by [71] and in Porto Alegre. Albeit small, the median duration of HWs also increased, resulting in an even higher number of days under the HW regime annually. These results corroborate the strong evidence that an increase in HW frequency and duration is widespread in SA from 1995 onwards, with a huge enhancement since 2000 [72]. Also, [73] found a significant rise in extreme temperatures over

Amazon, Northeast Brazil, and Southeast SA, with increasing trends of 1.20, 3.05, and 0.44 in the number of warm days per decade, respectively, evidencing the warming across the whole SA since the mid-20th century onwards. The highest tendencies observed in the present study were for the north, northeast, and central-western MRs, regions with less seasonal temperature variability. In particular, [4] estimated a considerable increase in hot days in SA over the 2006–2010 period, with significantly higher trends in regions closer to the equator. Moreover, a previous study conducted in six of the largest Brazilian cities (Manaus, Recife, Brasilia, Rio de Janeiro, São Paulo, and Porto Alegre) partially linked the increase in the HW frequency to the urbanization effect [71]. The urban heat island effect has great potential to amplify warm conditions and HWs within urban areas [74], and its occurrence has been reported in several Brazilian cities [75–79]. In particular, pervasive spatial and population growth has been noticed in all MRs analyzed here [80]. The Brazilian population observed a major urbanization process in the late 1970s when urban residents reached 56% of the population. This urbanization process kept its pace, and in 2000, this proportion surpassed 81% of the country's population [81]. The census that should be carried out in 2020 in Brazil is being carried out now. However, according to UN estimates, Brazil reached 87% urban population proportion in 2018 [47].

## 4.2. Twenty-first-century burden of HWs in Brazilian metropolitan areas

A large number of ED was attributed to the growing number of HWs that took place during the period analyzed in detail here (2000–2018), totaling 48,075 (40,448–55,279) heat-related ED, with a substantial portion occurring in the last five years. Such numbers, which are already alarming, could even worsen in the coming decades, considering the large increasing trends in hot temperature extremes and the population aging in these areas. Recently, a multi-country study by [21] (including 19 locations in Brazil) estimated that under an average global temperature increase of ~1°C, 37% of heat-related excess mortality across the regions studied can be attributed to additional heat exposure imposed by anthropogenic global warming. In particular, for the south and southeastern MRs, our results align with [32], which observed steep increases in mortality associated with rising temperatures in these areas, making them particularly vulnerable to extreme heat. Climate change projections point to a warmer scenario in the region defined by increasing levels in the mean temperature but also by strong nonlinear processes triggered by deep changes in soil moisture, precipitation, and evaporation regimes that will likely impose more frequent, intense, and prolonged HW episodes [4, 82–85]. This raises new challenges for the Brazilian authorities concerning protecting the ecosystems and public health. In parallel, the combined occurrence of droughts and HWs in some Brazilian regions seems to rise in recent years [37, 38].

The highly populated southeastern MRs presented the highest number of EDs, namely Rio de Janeiro and São Paulo. When excess mortality is normalized by population and by the number of days under the HW regime, larger mortality rates are observed in the MRs of Porto Alegre, Rio de Janeiro, and São Paulo, and particularly in Recife, Belém, and Cuiabá, which have experienced a large increase in recent years. The large mortality rates in the south and southeastern MRs of São Paulo, Rio de Janeiro, and Porto Alegre can be related to the high share of older people in the age pyramid in these regions. The fast increase experienced by the MRs of Belém, Recife, and Cuiabá can be interpreted in line with geographical inequalities in Brazil. In general, the less developed central-west, north, and northeastern regions have the worst health indicators compared to the south and southeastern, e.g., health care utilization and access, health resource supply, access to diagnosis, and, consequently, life expectancy, which varied directly with socioeconomic development [86–89].

## 4.3. The interplay of intersectionality and vulnerability to extreme heat in Brazil

Our results shed light on the relationships between demographic and socioeconomic heterogeneities and heat exposure in Brazil. The combined age classes of older and elderly people represented 75–94% of the ED, reinforcing that older people should be considered a priority group by the Brazilian health system during HWs. Larger percentages were observed in the south, southeast, and central-western MRs, which may be linked to the sharp North-South gradient in life expectancy. When analyzing the O/E ratios across both genders, females were more affected than males in all MRs, with more expressive disparities in Recife, Goiânia, Belo Horizonte, São Paulo, and Rio de Janeiro. In particular, some of these MRs have shown large gender gaps with poor gender parity indicators, such as on health for São Paulo and Recife, and on economic participation and opportunity, and political empowerment for the MRs of Goiânia and Belo Horizonte, according to [90]. Stratified assessment of heat-related mortality, with higher burdens among older people and females, has also been observed in several studies worldwide [62, 91–94]. Our findings can be analyzed in conjunction with risk estimates related to drought conditions in 13 Brazilian urban areas [52], which revealed females aged 65–74 years as the population group more vulnerable to the health effects of moderate, severe, and extreme drought.

Overall, age risk disparities can be partially explained by the higher vulnerability among older adults, mainly driven by their weakened immune system, lower resources, mobility and usually live isolates, high prevalence of chronic disease and comorbidities, and several physiological changes, such as reduced skin blood flow and sweating capacity, reducing the body's ability to dissipate heat, lower declining adaptive homeostasis and higher risk of dehydration [52, 95, 96]. Higher susceptibility among females is also in line with previous studies reporting gender inequalities in climate change risks due to societal, cultural, and economic factors, such as income inequalities, lower educational and employment status, and worse health status, among others [97]. Additionally, social conditions, such as living alone due to longer life expectancy compared to males, can also increase the susceptibility of elderly females.

Disparities in the risk of heat-related death associated with socioeconomic factors (race and education level) were also observed across the MRs. For males over 65 years, excess mortality for the low education sub-group was 10% (3%-18%) higher than for those with high educational level in the MR of São Paulo, 22% (12–33%) in Rio de Janeiro, 42% (17–71%) in Porto Alegre, 53% (30–80%) in Belém and 96% (56–147%) in Recife. Likewise, the excess risk for 65 years old and older females low-educational level subgroup was 8% (0–17%) in São Paulo, 24 (13–36%) in Rio de Janeiro, 36% (13–63%) in Belém, 62% (27–107%) in Recife, and 62% (16–128%) in Manaus, compared to low-education. Concerning race, the difference in excess mortality between black/brown people and white people was 10% (2–19%) in Rio de Janeiro, 15% (0–31%) in Belo Horizonte, 17% (9–25%) in São Paulo, 27% (0–60%) in Brasília, 33% (0–78%) in Manaus 40% (14–72%) in Recife, 44% (25–67%) in Belém, 65% (31–106%) in Goiânia, and 92% (17–213%) in Curitiba, for males with >65 years. For females (65 years and older), the results for the heat-related mortality increase among blacks and browns was 8% (2–16%) higher than for whites in the MR of Rio de Janeiro, 8% (1–15%) in São Paulo, 34% (11–62%) in Recife, 41% (22–63%) in Belém, 60% (25–104%) in Brasília, 82% (41–135%) in Goiânia, 203% (1–409%) in Cuiabá.

Educational level, as well as other socioeconomic aspects, are recognized as determining factors for health indicators such as morbidity, mortality, and life expectancy [98–101], playing a key role in relieving the levels of thermal stress during HW events (i.e. air-conditioned housing with good thermal insulation, easy access to quality healthcare, working indoors that does

not involve direct exposure to heat). In Brazil, such disparities have also been attributed to inequalities in access, use, and quality of healthcare services [88, 102, 103], particularly for the older adults [104]. Regarding racial disparities, it needs to be interpreted in line with well-documented evidence of racial inequities in health [105]. In Brazil, black and brown people are more likely to have a low-income status than white people [106], in addition to having unequal access to health care [107, 108].

The combined effect of educational level and racial disparities in heat-related health hazards corroborates with previous studies elsewhere [105, 109–112] and highlights the complex associations between climate changes, environmental justice, and environmental racism, which refers to the disproportional burden of environmental hazards among the lower socioeconomic and ethnic group [113–115].

## 4.4. Cause-specific attributable deaths during HWs

Overall, the analysis of leading causes of death suggested that heat-related ED are associated with several diseases, not only specific heat-related illnesses, indicating a harvesting effect, in which heat stress drives the death of chronically ill individuals in fragile health conditions. Among the main causes of death, diseases of the circulatory and respiratory systems and neoplasms were the most frequent. Higher risks for cardiovascular and respiratory deaths were also observed in a recent study in 326 Latin American cities [32]. Regarding circulatory diseases, [34] found increased cerebrovascular mortality for both hot and cold temperature extremes in Brazil. Similarly, [116] identified circulatory diseases as a leading cause of heat-related ED in Madrid, mainly related to rapidly fatal health outcomes, with a short time interval after exposure, often dying before hospital admission. Also, high blood pressure was a risk factor for the death of older people during the 2003 HW in France [112]. The increase in deaths from respiratory diseases during HWs is often related to the worsening of the conditions of people suffering from chronic respiratory diseases [117–119]. In most of these studies, a sustained rise in respiratory mortality was observed even after the end of the HW. Although this lag was not considered in the present study, the results also reflected a strong relationship between heat exposure and deaths from respiratory diseases. Finally, neoplasms have also been identified as one of the leading causes of heat-related death in Europe and Asia [118, 120–122], confirming that exposure to extreme heat may aggravate pre-existing problems of cancer patients.

Despite the lower absolute number of ED, diseases of the skin and subcutaneous tissue, diseases of the nervous system, and diseases of the genitourinary system presented high O/E in most MRs. Regarding skin and subcutaneous tissue, the main heat dissipation physiological mechanisms during thermal stress are sweat production, cardiac output increase, and blood flow redirection to the skin [123]. Therefore, delaying the thermoregulatory responses involved in heat dissipation can lead to skin disorders, such as dermatitis, psoriasis, ichthyoses, and ectodermal dysplasias [69]. One of the most critical functions of the autonomic nervous system is regulating body temperature [124], so distinct thermoregulatory disorders and illnesses have been related to heat stress [125]. Additionally, heat exposure has been shown to increase the burden for older people suffering from degenerative brain diseases, such as Alzheimer's [126] and Parkinson's disease [116]. Considering genitourinary diseases, a nationwide study in Brazil by [127] identified it among the health conditions more associated with HWs, in addition to skin problems and endocrine, nutritional, and metabolic diseases. According to the authors, dehydration, hypovolemia, and other clinical conditions associated with exposure to extreme heat can increase the burden of renal failure and other genitourinary impairments.

Mental and behavioral disorders also presented a substantial increase in mortality rates in some MRs, similar to those observed in Europe [93, 112]. In particular, a study conducted in Australia by [128] identified a positive association between high temperatures and hospital admissions/mortality for elderly people with schizophrenia, schizotypal, delusional disorders, and dementia. In addition, people with mental illnesses have, on average, a lower socioeconomic status, fragile health, and live in complete social and familiar isolation. These are all risk factors for heat-related mortality.

Although uncertainties for most causes of death associated with the O/E values were smaller than 20%, certain ICD categories presented larger confidence intervals. Notably, ICD XII (Diseases of the skin and subcutaneous tissue) had uncertainties exceeding 90% of the O/E value in the MRs of Goiânia, Porto Alegre, Recife, Curitiba, and Brasília, besides O/E values indistinguishable from unity in the MRs of Manaus and Cuiabá. Similarly, ICD V (Mental and behavioral disorders) exhibited uncertainties above 40% of the O/E value in most MRs. For ICD V, excess deaths during HWs were not statistically significant in the MRs of Cuiabá, Fortaleza, Goiânia, and Manaus. The low total mortality counts for these specific diseases certainly influence large uncertainties for ICD V and XII, indicating the need for long-term studies focused on these specific causes in Brazil. Despite substantial uncertainties, excess mortality for these particular ICD chapters was statistically significant in some MRs, especially those with larger populations, and, thus, greater statistical significance. Regarding individual MRs, uncertainties in the O/E values were below 20% for most ICDs, except for the MR of Cuiabá, where uncertainties exceeded 25%, yielding non-significant results for ICDs XII and V, but also for ICD XIV (Genitourinary system diseases). The small population and the lower number of HWs compared to the other MRs can partially explain large uncertainties for the MR of Cuiabá.

It is important to point out that the prevalence of chronic diseases has been shown to be partially driven by social and demographic inequalities in Brazil. According to [129], the elderly, women, low educational level, black, brown, and indigenous people, among other factors, are more susceptible to multiple chronic conditions, in particular hypertension, one of the leading risk factors for cardiovascular diseases (one of the leading causes of heat-related ED in this study). In this context, the combination of the expansion of primary health care and the reduction of income inequalities has positively affected mortality and morbidity indicators in Brazil [130].

## 4.5. Rising temperatures represent an emerging public health challenge in Brazil

When we looked at the HWs with the highest increase in ED by MR, most were short (3–5 days) and low-intensity events. In some of the central-west and northern MRs (Belém, Cuiabá, and Brasília), these HWs have occurred in the winter/dry season. However, in the south and southeast, they occurred mostly in the summer, indicating that warning systems must be regionally adapted to the time of year. It is also interesting to notice that most highly impacting HWs occurred just 1–2 weeks after a previous HW (e.g., in the MRs of Belém, Recife, Porto Alegre, Florianópolis, and Rio de Janeiro). This result can be partially interpreted in line with the EHF definition, considering that if a HW is preceded by another HW within an interval of less than 30 days, this implies a lower value of the EHF acclimatization index, which results in a HW classified as low intensity according to the definition. However, in this case, the acclimatization effect does not happen smoothly. On the contrary, a temperature spike of a few days is responsible for a rise in the 30-day temperature average (used in calculating the EHF acclimatization index). In addition, recent studies have shown that the impacts of consecutive hazards can significantly weaken the health situation, as there is a greater chance of increasing the

strain on both the health service and the population's health due to repeated extreme heat. The prevalence of these consecutive disasters is increasing due to climate change and socioeconomic exposure and vulnerability [131]. This contextual information is important to help explain how some of these relatively short-term events induced such a large impact in terms of excessive mortality. As HWs are mostly short-term events leading to fast stress on health services, these events require quick responses, which need to be adapted to the regional risk profile of the population according to social and demographic aspects and clinical conditions. The predominance of low-intensity HWs among the events with the largest increase in mortality also indicates that the use of EHF for classifying HWs in terms of intensity may not be adequate for the regional characteristics in Brazil, which certainly motivates future research on the construction of new indices that better represent the impact of heat on the health of the population in Brazilian areas.

The results presented here also point to the need to integrate the discussion on the health impacts of HWs into the Brazilian Unified Health System (SUS) to prevent heat-related hospitalization and death, but also the stress in primary health care services and health professionals, as discussed by [132]. Conversely, no one of the high-impact HWs was captured through the identification of rumors performed with EIOS, despite the observed increase in the publication of HW-related news in the media in recent years in Brazil, which underlines the lack of visibility of the problem. In this sense, it is worth improving the sensitivity of the capture of HWs to subsidize the health surveillance process and the adoption of health risk management measures. It also highlights the need for warning community systems and emergency actions in response to extreme weather conditions, a particular challenge in Brazil, considering structural problems of the SUS, such as organization and governance gaps, underfunding, and significant regional disparities [86, 133], as observed in the period of the COVID-19 pandemic [134].

## 4.6. Strengths, drawbacks, and future work

The burden of heat-related excess mortality has been assessed in recent studies worldwide, confirming a relevant impact of population exposure to high temperatures on human health. However, most of these studies have focused on a few diseases, such as cardiovascular and respiratory diseases. In addition, the risk profile of different population subgroups according to social and demographic aspects is still a topic that has been explored little in these works. Thus, one of the key strengths of this study is that it investigates the role of socioeconomic and demographic factors on the vulnerabilities related to heat exposure and disaggregates the impact according to specific causes of death, many of them little explored in the literature. Assessing the joint influence of demographic and social drivers on the vulnerability profile at the regional level is valuable information for adaptation strategies and responses from the public health system, particularly in densely populated and socially stratified urban areas, and for the construction of regional risk maps. Still, there are some limitations in this work that we must acknowledge, particularly regarding the relatively short span of the mortality data (only from 2000 onwards), the limited number of socioeconomic indicators available on DATASUS, and the use of meteorological data based on a single weather station for each MR. Further works could also extend the approaches developed here to include more demographic dimensions, hospital admission data, estimations of economic costs, and other meteorological data than surface air temperature (e.g., humidity). Another issue is the absence of analyses of sex differences stratified by age. Although we observed higher mortality rates among women, it is hard to quantify whether this is due to the women/men survival bias or due to biological or environmental differences in exposure or response to heat. Since policy implications are huge, we recommend further investigation into this aspect.

Moreover, our methods are not able to measure the influence of events such as droughts and air pollution episodes, or even epidemics caused by arboviruses, such as dengue, on excessive heat-related mortality [135] whose magnitude of outbreaks shows a short-term association with the increase in temperature in several regions [136]. The occurrence of compound extreme events has been shown to strengthen heat-related mortality risk, particularly those that mix drought, HWs, and wildfires [8, 12, 137, 138]. Also, a previous study in Portugal found an amplification of heat-related mortality during the COVID-19 pandemic [14] due to healthcare system disruptions and lower attendance by the population in healthcare facilities. The combined effects of extreme temperatures and epidemics are particularly worrying in Brazil, where COVID-19 has caused a high burden on mortality. As discussed before, a limitation of this paper is that it does not include any lags in the effects of heat on mortality, particularly since other studies have demonstrated mortality effects during 1–3 days after exposure [139]. As such, our approach can underestimate the mortality effects and suggest that the number of heat-related excess deaths can be higher in most of the MRs. We point out, however, that the EHF calculation considers a three-day-averaged daily mean temperature and somehow intrinsically adds a more extended period to the HW. Finally, we are aware that, despite the overall relevance of all the considered MRs in this study, it remains a fact that they just cover circa 35% of Brazil's population (74 million people), most of them in the southeast (54%), so extrapolation of their representativeness to the country level should be done with caution. Furthermore, although the urban population is more exposed to higher and more frequent HWs, previous studies have reported higher heat-related mortality risks in rural areas [140]. Urban–rural differences can be related to differences in age structure, occupation types, level of education, and access to healthcare services and air conditioners, which certainly needs to be investigated in Brazil. The differences in health care in urban and rural areas in Brazil can be quite large, mainly due to the travel distances [141]. Despite these caveats, this study presents one of the most comprehensive analyses of the HW impacts on mortality in Brazil, in some of the most densely populated regions of the country, providing innovative insights into the climate change and human health interface.

## 5. Conclusions

Increasing trends in the frequency, intensity, and duration of the HWs were observed in all MRs analyzed, taking a toll on a significant fraction of the population in terms of exposure to extreme heat in Brazil over recent years. However, results revealed that females, elderly, black and brown people, and those with lower educational levels are the most affected population sub-groups, highlighting how anthropogenic-induced climate change has already exacerbated socioeconomic inequalities in Brazil. These findings evidence that socioeconomic and demographic aspects are important factors to be considered by policy-makers in Brazil for the development of HW early warning systems and adaptation plans to emerging impacts of climate change on human health, taking into account the vulnerability and resilience of multiple population sub-groups. Moreover, suffering from a variety of pre-existing diseases was found to be an important risk factor that contributes significantly to heat-related excess mortality, being, therefore, a relevant aspect that should be taken into account when designing public health prevention strategies. Although the excess mortality estimated in this study (48,075 ED) was 20 times greater than the number of deaths associated with landslides nationwide [142] in the same period, extreme heat is still a neglected disaster in Brazil, as pointed out by the results of our event-based surveillance analysis. This scenario is further worsened by the growing population aging experienced by the Brazilian population in the last decade, according to the IBGE 2022 Census (people aged 60 or over rose from 11.3% to 14.7% of the population between

2012 and 2023). Our results indicate that expanding primary health care combined with reducing socioeconomic and gender inequalities, e.g., implementing effective social policies, represents important steps to reduce heat-related deaths in Brazil. It is of utter importance to highlight that under any future global warming scenario, even fulfilling the most rigorous Paris Climate Agreement targets, the impacts of HW on the human population are expected to increase, and the health service needs to be prepared, particularly for vulnerable groups and people suffering from comorbidities. This is a notable challenge in Brazil, considering the structural problems of the Unified Health System, such as organization and governance gaps, low public funding, and significant regional disparities.

## Supporting information

**S1 Fig. Monthly frequency of occurrence of heat waves for low-intensity (blue) and severe/ extreme (red) for the 14 Metropolitan Regions (MRs) analyzed.** The Northern MRs are presented in the first row, Northeastern MRs are in the second row, Central-western MRs are in the third row, Southeastern in the fourth row, and Southern MRs are presented in the fifth row.
(TIF)

**S2 Fig. Distribution of all-natural deaths in the 14 Metropolitan Regions (MRs) analyzed between 2001 and 2018.** The percentage of death among older and eldest is 47.65% in the MR of Manaus, 47.99% in Cuiabá, 48.58% in Salvador, 49.92% in Belém, 51.16% in Brasilia, 51.93% in Goiânia, 54.87% in Recife, 55.57% in Belo Horizonte, 55.82% in Curitiba, 56.90% in Fortaleza, 57.06% in São Paulo, 57.65% in Florianópolis, 59.65% in Porto Alegre, and 59.90% in Rio de Janeiro.
(TIF)

**S1 Table. Observed-to-expected ratio (and confidence interval) and total number of excess deaths (ED) for all HWs identified in the MRs for ICD-10 chapters.** ICD-10 chapters: II (Neoplasms), IV (Endocrine, Nutritional and Metabolic Diseases), IX (Diseases of the Circulatory System), V (Mental and Behavioral Disorders), VI (Diseases of the Nervous System), X (Diseases of the Respiratory System), XI (Diseases of the Digestive System), XII (Diseases of the Skin and Subcutaneous Tissue), XIV (Diseases of the Genitourinary System).
(PDF)

## Author Contributions

**Conceptualization:** Djacinto Monteiro dos Santos, Renata Libonati, Beatriz N. Garcia, João L. Geirinhas, Leonardo F. Peres, Ana Russo, Renata Gracie, Helen Gurgel, Ricardo M. Trigo.

**Data curation:** Djacinto Monteiro dos Santos, Renata Libonati, Beatriz N. Garcia, João L. Geirinhas, Renata Gracie, Ricardo M. Trigo.

**Formal analysis:** Djacinto Monteiro dos Santos, Renata Libonati, Beatriz N. Garcia, Barbara Bresani Salvi, Eliane Lima e Silva, Ricardo M. Trigo.

**Funding acquisition:** Renata Libonati, Renata Gracie, Helen Gurgel.

**Investigation:** Djacinto Monteiro dos Santos, Renata Libonati, Beatriz N. Garcia, João L. Geirinhas, Barbara Bresani Salvi, Eliane Lima e Silva, Leonardo F. Peres, Ana Russo, Renata Gracie, Helen Gurgel, Ricardo M. Trigo.

**Methodology:** Djacinto Monteiro dos Santos, Renata Libonati, Beatriz N. Garcia, João L. Geirinhas, Barbara Bresani Salvi, Eliane Lima e Silva, Ana Russo, Renata Gracie, Helen Gurgel, Ricardo M. Trigo.

**Project administration:** Renata Libonati, Renata Gracie, Helen Gurgel.

**Resources:** Renata Libonati, Renata Gracie, Helen Gurgel.

**Software:** Djacinto Monteiro dos Santos.

**Supervision:** Renata Libonati, Ricardo M. Trigo.

**Validation:** Djacinto Monteiro dos Santos, Ricardo M. Trigo.

**Visualization:** Djacinto Monteiro dos Santos, Julia A. Rodrigues, Ricardo M. Trigo.

**Writing – original draft:** Djacinto Monteiro dos Santos, Renata Libonati, Beatriz N. Garcia, João L. Geirinhas, Barbara Bresani Salvi, Eliane Lima e Silva, Ana Russo, Renata Gracie, Helen Gurgel, Ricardo M. Trigo.

**Writing – review & editing:** Djacinto Monteiro dos Santos, Renata Libonati, Beatriz N. Garcia, João L. Geirinhas, Barbara Bresani Salvi, Eliane Lima e Silva, Julia A. Rodrigues, Leonardo F. Peres, Ana Russo, Renata Gracie, Helen Gurgel, Ricardo M. Trigo.

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
