## [Decision Letter · Decision Letter 0]

13 Jul 2023

PONE-D-23-10924Twenty-first-century demographic and social inequalities of heat-related deaths in Brazilian urban areasPLOS ONE

Dear Dr. Monteiro dos Santos,

Thank you for submitting your manuscript to PLOS ONE. After careful consideration, we feel that it has merit but does not fully meet PLOS ONE’s publication criteria as it currently stands. Therefore, we invite you to submit a revised version of the manuscript that addresses the points raised during the review process. After carefully reading the manuscript and the comments sent by the reviewers, we would like to invite you to provide a major revision to your text in order to be reconsidered for publication. The three reviewers agreed on the relevance of the discussion proposed, but both also agreed on the need for long corrections. As a summary, we suggest a review of the general framework, situating better your debate about what 'rumor' exactly means and giving more details on how the data can be used, and also the question about all deaths and injuries that all reviewers indicated. As well as a further inquiry into the state of arts of studies in Latin America.

Reviewer 1 and 2 indicates issues concerning the age, sex, and race variables that must be addressed. Please take them into account as they may lead the article from what seems to be a good but fragile argument to a good and strong piece of literature. Thank you for submitting your article to PLOS ONE. We hope these recommendations, together with the detailed comments from the reviewers, will contribute to the improvement of your work.

We look forward to receiving your revised manuscript.

Kind regards,

Ivan Filipe de Almeida Lopes Fernandes, Ph.D.

Academic Editor

PLOS ONE

“D.M.S. acknowledges the support of FIOCRUZ [grant VPPCB-003-FIO-19] and FAPERJ [grant E-26/205.890/2022]. RL was supported by FAPERJ [grant E26/202.714/2019] and CNPQ [grant 311487/2021-1]. A.R. and R.M.T. were supported by Fundação para a Ciência e a Tecnologia, I.P./MCTES through national funds (PIDDAC)” –UIDB/50019/2020 and also by Project ROADMAP (JPIOCEANS/0001/2019). B.N.G. was supported by CNPQ [grant 161075/2021-5]. J.L.G. acknowledges the support of FCT (Fundação para a Ciência e Tecnologia) for the PhD Grant 2020.05198.BD**.**”

Additional Editor Comments:

After carefully reading the manuscript and the comments sent by the reviewers, we would like to invite you to provide a major revision to your text in order to be reconsidered for publication. The three reviewers agreed on the relevance of the discussion proposed, but both also agreed on the need for long corrections. As a summary, we suggest a review of the general framework, situating better your debate about what 'rumor' exactly means and giving more details on how the data can be used, and also the question about all deaths and injuries that all reviewers indicated. As well as a further inquiry into the state of arts of studies in Latin America.

Reviewer 1 and 2 indicates issues concerning the age, sex, and race variables that must be addressed. Please take them into account as they may lead the article from what seems to be a good but fragile argument to a good and strong piece of literature. Thank you for submitting your article to PLOS ONE. We hope these recommendations, together with the detailed comments from the reviewers, will contribute to the improvement of your work.

Reviewers' comments:

Reviewer's Responses to Questions

**Comments to the Author**

1. Is the manuscript technically sound, and do the data support the conclusions?

Reviewer #1: Yes

Reviewer #2: Partly

Reviewer #3: Yes

2. Has the statistical analysis been performed appropriately and rigorously? 

Reviewer #1: Yes

Reviewer #2: I Don't Know

Reviewer #3: Yes

3. Have the authors made all data underlying the findings in their manuscript fully available?

Reviewer #1: Yes

Reviewer #2: No

Reviewer #3: Yes

4. Is the manuscript presented in an intelligible fashion and written in standard English?

Reviewer #1: Yes

Reviewer #2: Yes

Reviewer #3: Yes

5. Review Comments to the Author

Reviewer #1: The manuscript is well written , tackling an important issue about heatwaves and mortality in the Brazilian major cities.

However, it can be improved, following some suggestions given below:

First of all, why all diseases? Many of the cited diseases are rather complex with indirect weather impact, difficult to establish any correlation. Infection diseases are one of them where vectors and bacteria/viruses are involved. I strongly suggest modifying it, keeping the main diseases supported by literature such as respiratory, cardiovascular and maybe neoplasm. The latter is also complex since, for example, cancers caused by UV and stronger radiation, during HW, can be the triggering but not the final death. Heat waves have a short duration to present these impacts on death tolls.

Minor comments.

The last paragraph of the Introduction section, starting with 'Accordingly', should be separated in a new paragraph.

São Paulo City means Metropolitan Area of Sao Paulo, not only the city itself. The same for Rio.

Black/white dichotomy: Brazilian population is rather blended. According to Pena et al (2009), the 'racial' mixture is so high that cannot be used for medical proposals. Therefore, add a paragraph about it, since the results are interesting.

equation 1. explain Ti. Why did not use max temperatures? I also suggest modifying it to include that variable. Mean temperature smooths a lot the heat wave impacts.

Mij is the observed number of deaths, right? Add it.

Explain what 'rumor' exactly means and give more details how the data can be used in section 2.5.

ED popped out without a proper introduction. It means expected deaths, right?

During the sections of Results there are many 'highly than' and similar without any values to support such statements. Or at the end of 3.5 section without percentiles. Please provide.

The last sentence of section 3.6: 'none rumors were found', that means no media shows that? Please clarify.

Discussion.

Please provide a range of the HW trends among the cities. The results show different behavior.

'Significant rise of ...' gives numbers and an explanation why, since the climate is already hot.

2006-2100 is spelled wrongly: 2006-2010.

Another point that should be stressed. All data is based on deaths per 100,000 or so inhabitants , right? It is a percentile not the absolute number because the population in each category is different.

Despite the fact that the authors cited humidity as a gap in the manuscript, I also suggest including it (see Fernanda Diniz thesis, 2022), because high humidity can overload the elderly metabolic capacities.

Figure 6 displays another source.

Reviewer #2: PONE-D-23-10924: Twenty-first-century demographic and social inequalities of heat-related deaths in Brazilian urban areas

This article focuses on exploring demographic and socioeconomic differentials in susceptibility to the effects of heat waves in 14 Large Brazilian cities. The authors use an index to quantify the duration and intensity of heat waves, and then calculate a measure of excess mortality during these heat waves. They find heterogeneity in the impacts of heat wave by city, and stronger effects among the elderly, women, racial/ethnic minorities and individuals of low educational level. I have a few concerns I have outlined below:

Major comments:

1. Introduction: the authors frame the need for their study around a lack of studies exploring heterogeneities in the effects of heat on health, especially in Latin America (and in Brazil in particular). First, I’d like to point out is that the Kephart 2022 study they cite does explore regional heterogeneity within Brazil, as it shows results of rate ratios for each Brazilian city, showing how tropical areas are generally less sensitive to heat (due to lower temperature variation). This other paper by the same group (https://www.sciencedirect.com/science/article/pii/S0277953622008322) looks at social and demographic characteristics of cities in relationship with these heterogeneities. Second, I think the authors should cite more comprehensive evidence regarding the lack of evidence in Latin America. A good paper to cite may be this one by Berrang-Ford (https://www.nature.com/articles/s41558-021-01170-y) where the authors find a lack of research in the region.

2. Setting: how are MRs defined? I assume there’s an IBGE definition, but this is not mentioned.

3. Data: the authors indicate some percent of missing values, but it isn’t clear whether this corresponds to temperature or mortality. In any case: what’s the pattern of this missingness? Is it more common in warmer months? It is unclear how the missing data was handled. Did authors conduct a complete case analysis, excluding all missing data?

4. Mortality data specifically: what’s the justification for excluding injuries? There’s some research showing increases in homicides with heat, for example (https://pubmed.ncbi.nlm.nih.gov/28687898/). If the missing data %s in the data section correspond to temperature, what’s the missingness in mortality data? I know education and race tend to have higher %s of missing data. Speaking of race: what happens with yellow and indigenous groups? (as I think they are called in Brazil) They seem to be excluded from this article (I understand their sample size may be small, but I’d mention explicitly if these are excluded deaths). Another potential source of missing data are ill-defined deaths or garbage codes, which the authors don’t seem to be addressing (and which is important especially if the number of these types of deaths changes in heat waves).

5. Exposure: The authors use what seems to be like a widely used metric of heat waves, the excess heat factor or EHF. This is calculated based on two indices: (1) a significance index, which is the difference between a 3-day running mean of daily temperature vs the 95th percentile of historical (between 1981-2010—why specifically those years?) temperature over the whole year; and (2) an acclimatization index, which is the difference between this 3-day running mean and the 30-day mean temperature. Both indices are then multiplied, taking the raw significance index and modifying it by the acclimation index. This EHF, if positive, indicates a heat wave, and is then compared to all positive EHFs (I assume historically? For the same city? Unclear from the text), and if it falls below the 85th percentile, the heat wave is considered low intensity, if it falls above 3x the 85th percentile it’d be extreme, and otherwise it’d be severe. If this explanation is incorrect, there may need to be some clarification done in the text. What is the reasoning behind differentiating between severe and extreme heat waves if they were interpreted as a single category?. Furthermore, is there any temporal restriction conducted here? For example only considering summers for the cities where this makes sense (non equatorial)? How are cold temperatures considered? Does this index already take that into account, since the EHF would be negative in those cases?

6. Excess mortality estimation: in calculating the O/E ratio, the authors compare actual mortality during each heat wave (Mij) to a historical average in previous and subsequent years. But is this average conducted using the same days of the year? And what do the authors mean by “Reference periods under HW condition, either in an earlier or later year, were not included in the calculation of expected mortality.”? Does this mean that if a day fell under a HW, they wouldn’t include it here? Last, there’s no mention to what software was used for all statistical analysis.

7. Rumor search: what is the purpose of this section in the methods section? (2.5). How is a ‘rumor defined?I don’t see this mentioned in the objectives or justified in the introduction, and it doesn’t seem to be connected to the rest of the analysis (at least insofar one can find this information in the methods section). This seems to be used in section 3.6 to identify whether the most impactful heat waves were actually picked up by media. However, while an interesting finding, this is seems to be out of the scope of this paper, and would need way more detail, justification, statistical testing, etc, to be part of these analysis in my opinion.

8. Descriptive results: Figure 2 is great, see some comments in the minor comments section. I think table 2 would benefit from adding the confidence intervals to the slope, instead of underlining the nonsignificant ones (if I understand the footnote correctly). Moreover, this slope analysis does not seem to be described in the methods section, so it is unclear to me how the trend was calculated. Also, the headings on Table 2 are a little unclear. Why were the frequency of HW events assessed through ‘20s (does this include through 2022?) while the duration of HWs only through ‘10s. Why was the decade of the 1990’s completely skipped over? The authors may want to be more specific and just indicate the ranges (e.g. 2000-2020 instead of “2000’s/2010’s”)

9. Main results: table 3 and figure 3 outline the main results. I suggest the authors sort this table and figure in the same way as Table 2 and Figure 2, to not confuse the reader. The figure shows the O/E ratios (although it’s missing the confidence intervals, which would be tricky to show here, but which could be shown in a similar figure as Figure 6 here (https://read.dukeupress.edu/demography/article/55/4/1363/167907/Bayesian-Estimation-of-Age-Specific-Mortality-and). In table 3, it’d be great to see confidence intervals around the excess deaths to know how much variability there’s around these estimates. Last, for figure 3, since the y-axis is a ratio, it should be plotted in the log scale (given that ratios can go from 0 to infinity, and therefore the distance from 0 to 1 should be the one as from 1 to infinity [or, to use a more realistic example: distance of 0.5 and 1 should be the same as 1 to 2]).

10. Stratified results: Figure 4 and 5 show the results stratified by age, sex, education and race, showing that elderly people, women, and people of low education and black/brown people have higher ED due to heat. A few comments: (1) the first part of Figure 4 is hard to interpret without normalizing by the number of people in each age group; (2) for both the second part of figure 4 and all of figure 5, I wonder if the authors could compute a ratio between women/men, black/brown and white, high and low education, to know what the excess risk is (And confidence intervals around those ratios); (3) as with Figure 3, both figure4b and figure 5 should be plotted in the log scale (see explanation above).

11. Results by cause of death: this section would greatly benefit from considering confidence intervals. I see a few causes of death that are very rare (e.g. skin and subcutaneous tissue) with very high O/E ratios, potentially because of very low mortality rates to begin with. By addressing significance, or at least acknowledging variability around the O/E, the authors can focus on the more meaningful results. Additionally, authors should consider defining what ‘not elsewhere classified’ means. Is this related to missing data or ill-defined codes? What % are not classified?

12. The results by specific heat wave (table 4) are interesting, but have not been detailed in the methods section. Again, confidence intervals would help here, along with a normalization of deaths by population (the high numbers of Sao Paulo and Rio de Janeiro may be just because of the size of these cities)

13. The authors have written a good discussion, contextualizing results with prior studies and providing what seems to be adequate interpretation. The only part I think I’m missing is a more methodological discussion comparing the results to articles that look at temperature overall (instead of heat waves, for example using distributed lag non linear models), those that look at cold vs hot temperatures, etc. It may be helpful to understand the rationale for the absence of a lag period in the analysis. Additionally, discussion may benefit from addressing why most of the excess mortality events occurred during low-intensity heat waves rather than during severe/extreme heat waves.

Minor comments:

14. Abstract: the abstract points to both older folks/women and low educational level/black/brown people having a higher susceptibility to heat, but the sentence where this is shown makes it seems like these correspond to two different analyses. Are they part of the same result?

15. Figure 1: while I appreciate the very complete descriptive figure, the Koppen climate figure is a bit confusing, as it includes all of South America, with unclear borders of where Brasil is. My intuition is that the thicker lines are Brazilian states, but since they aren’t showed anywhere else in this figure it’s hard to know where Brazil ends in this figure. As it is, it gives the impression of wider climate heterogeneity (as it includes the high altitude regions of Peru, Bolivia and Chile, and the desert regions of Chile and Argentina. Furthermore, it isn’t clear to me what the pink bars are (they look to be bars indicating population, but we lack axes, and the information of population is already there).

16. Figure 2: I commend the authors for sorting the cities meaningfully (by latitude). However, it’d be great to include a label that makes this obvious to the reader. For example, since Belem is around the equator, you could include a line next to Belem and label it Equator, with an arrow pointing up (North) and one pointing down (South). In any way, this is an excellent figure! It shows increasing number of heat waves and intensity, especially in the South.

Reviewer #3: The analysis describes long-term trends and excess deaths from heat waves in 14 large metropolitan areas in Brazil over a period of decades. While other studies have examined the impacts of hot temperatures on mortality, this analysis helpfully provides findings on the impacts of consecutive hot days (i.e. heat waves), which plausibly have distinct, cumulative effects on physiological stress and death. The analysis of variability in excess deaths by sex, age, race, and educational attainment is important and useful for policymakers. The paper could be improved by addressing the following points of confusion and by acknowledging some major limitations of the analysis which are not sufficiently described.

Major comments

-The analysis excludes deaths from external causes (e.g. accidents and homicide), but the authors’ describe the analyzed deaths as “all-cause” throughout the manuscript. It is more common to call non-accidental deaths “non-injury deaths” or “natural deaths” or similar, instead of calling them all-cause deaths. All-cause deaths typically mean truly all causes, without excluding deaths from external causes.

-The authors do not sufficiently discuss the interaction between age and gender in which subpopulations have increased risk. Older adults have greater risk, and women have greater risk, however, we know that older adults are disproportionately women because a greater proportion of men have already died. The manuscript briefly mentions that elderly women may be more likely to live alone and therefore have higher risk, but more attention needs to be given to the large sex differences in the age distribution of the population, particularly at older ages. If the authors could add analyses of sex differences stratified by age that would be helpful. If not, they should soften the description of the findings on sex disparities and add discussion of the limitation of not knowing whether this is due to survival bias or due to biological or environmental differences in exposure or response to heat. The policy implications are large, so this is an important nuance to present correctly.

-The modeling approach (as described) does not include any lags in the effects of heat on subsequent mortality. This is a major limitation of the paper, as other analyses from around the world have consistently demonstrated mortality effects 1-4 days after heat exposure. Given this major limitation, it is likely that the mortality effects described in this manuscript are underestimates of the true mortality effects, since deaths on subsequent days are not considered. This deserves a notable discussion in the manuscript.

-The authors hypothesize in Section 4.5. that some low intensity HWs make have had high impact because they occurred a few weeks after other HW events. However, wouldn’t this conflict with the assumptions in the HW assimilation metric, which assumes that heat within the past 30 days has an assimilating effect on the population?

-I did not find Figure 3 to be useful. Consider moving to the supplement or removing. Since this focus is on cumulative risk, not isolating specific events, the jittered dots of individual HW events are not very informative to the reader. There is also no real need to have the dual-axis plot. Consider breaking into two panels.

-Education variables: 1) the Methods describe two categories, low and high, however these do not seem to include the entire range of educational attainment. How were these categories chosen and what happened to individuals who fell between the low and high education categories? It is not clear. 2) Was this data included in the individual death records or is this area level data based on residence?

-Temperature data: Please comment on general patterns of where the meteorological stations were located within the metropolitan regions. In the urban core? Airports? Outside of the city? This is important for readers to assess how well the single monitor in each metro area may reflect true population exposures.

-Race: please justify the choice of combining black and brown into one category. The authors describe how three groups are often used in research in Brazil, but the final variable appears to be a binary.

-Supplement: I did not receive a supplement, though it is referenced in the manuscript.

6. PLOS authors have the option to publish the peer review history of their article (what does this mean?). If published, this will include your full peer review and any attached files.

Reviewer #1: **Yes: **FÁBIO LUIZ TEIXEIRA GONÇALVES

Reviewer #2: No

Reviewer #3: No

---

## [Author Response · Author response to Decision Letter 0]

26 Aug 2023

In order to respond to reviewer and editor comments in a more fluid way, connecting the points raised by each reviewer, new references and figures, we opt to send a separate letter with the title "Answer to reviewer comments" available in the attached files.

---

## [Decision Letter · Decision Letter 1]

16 Oct 2023

PONE-D-23-10924R1Twenty-first-century demographic and social inequalities of heat-related deaths in Brazilian urban areasPLOS ONE

Dear Dr. Monteiro dos Santos,

Thank you for submitting your manuscript to PLOS ONE. After careful consideration, we feel that it has merit but does not fully meet PLOS ONE’s publication criteria as it currently stands. Therefore, we invite you to submit a revised version of the manuscript that addresses the points raised during the review process. 

The reviewers have accepted most of the feedback given in the first revision of the manuscript. Nonetheless, they point out important issues that should be addressed. I would therefore request that you provide a detailed response to the comments made by the reviewers in order that the manuscript can be published in PlosOne. 

We look forward to receiving your revised manuscript.

Kind regards,

Ivan Filipe de Almeida Lopes Fernandes, Ph.D.

Academic Editor

PLOS ONE

Journal Requirements:

Additional Editor Comments:

Dear authors,

The reviewers have accepted most of the feedback given in the first revision of the manuscript. Nonetheless, they point out important issues that should be addressed.

would therefore request that you provide a detailed response to the comments made by the reviewers in order that the manuscript can be published in PlosOne.

Regards,

Reviewers' comments:

Reviewer's Responses to Questions

**Comments to the Author**

1. If the authors have adequately addressed your comments raised in a previous round of review and you feel that this manuscript is now acceptable for publication, you may indicate that here to bypass the “Comments to the Author” section, enter your conflict of interest statement in the “Confidential to Editor” section, and submit your "Accept" recommendation.

Reviewer #1: All comments have been addressed

Reviewer #2: (No Response)

Reviewer #3: All comments have been addressed

2. Is the manuscript technically sound, and do the data support the conclusions?

Reviewer #1: Yes

Reviewer #2: Partly

Reviewer #3: Yes

3. Has the statistical analysis been performed appropriately and rigorously? 

Reviewer #1: Yes

Reviewer #2: No

Reviewer #3: Yes

4. Have the authors made all data underlying the findings in their manuscript fully available?

Reviewer #1: Yes

Reviewer #2: No

Reviewer #3: Yes

5. Is the manuscript presented in an intelligible fashion and written in standard English?

Reviewer #1: Yes

Reviewer #2: Yes

Reviewer #3: Yes

6. Review Comments to the Author

Reviewer #1: The overall manuscript is almost ready for publication.Minor suggestions/reviews should be addressed as it follows:

1. Concerning races. Currently, it is no longer reliable to classify diseases by 'race' in hospital discharges, given the great homogeneity of our species (Tishkoff & Kidd,2004, and Lorusso & Bachini, 2015). D. Roberts (2011) and J Dupré (2008) emphasize that the biological term for race is a form of racism, and it is a mistake to accept that races, in the plural, exist.

The Genome Project even states the following:The completion of the Human Genome Project in 2003 confirmed humans are 99.9% identical at the DNA level and there is no genetic basis for race. Pena (2005) presented that only 0.0005 to 0.001% of our genome is characterized by the phenotype (the correct word for those 'differences', rather than 'races').

therefore I suggest that you keep your finds but use the above statements to explain the readers.

2. In the eq. (1) show that Ti is corresponded to day 1, and i+1 to day 2 etc.

3- pg 15 'small time interval'? How small? The same for 'significant rise' at pg 16.

4- Pgf from pg 18 is too long. Do a new one after the word 'overall'.

5- pg 21 HW have occurred in winter ... Add winter/dry season. There is no 'winter' there.

Reviewer #2: Response to Authors – PLOS one

We thank the authors for their changes to the paper. We still have a few outstanding issues that have not been addressed from our prior comments:

[R2-3] – Thank you for addressing some of our comments. Please provide clarity regarding whether a complete case analysis was done, and the potential impact for the cities with high missingness of temperature data (e.g., Florianapolis).

[R2-4] –Thank you for addressing some of our comments. Please clarify what is meant by non-natural confounding variables. Additionally, it is still unclear how missingness of demographics was ‘carefully addressed in the O/E ratio’? Clarify what was done with deaths missing race and education data – if they were excluded, please address potential biases (e.g., selection bias). Lastly, please describe how ill-defined deaths vary during heatwaves.

[R2-6] – Thank you for addressing our comments. Please address which software was used for all analyses, not just the slope analysis

[R2-7] – Thank you for addressing our comments. Please consider adding the purpose of the event-based surveillance analysis (rumor search) to the objective of the study.

[R2-11] –Thank you for including Table S1, but please interpret the uncertainty around the coefficients in the results (e.g., ICD XII has very wide confidence intervals). There are very wide confidence intervals in some of the ICD chapters.

Reviewer #3: (No Response)

7. PLOS authors have the option to publish the peer review history of their article (what does this mean?). If published, this will include your full peer review and any attached files.

Reviewer #1: **Yes: **FÁBIO LUIZ TEIXEIRA GONÇALVES

Reviewer #2: No

Reviewer #3: No

---

## [Author Response · Author response to Decision Letter 1]

17 Nov 2023

We respond to specific reviewer and editor comments in the document "PONE-D-23-10924 - Answer to reviewer comments #2", uploaded as Response to Reviewers in the Attach Files

---

## [Editor Report · Decision Letter 2]

29 Nov 2023

Twenty-first-century demographic and social inequalities of heat-related deaths in Brazilian urban areas

PONE-D-23-10924R2

Dear Dr. Santos, 

We’re pleased to inform you that your manuscript has been judged scientifically suitable for publication and will be formally accepted for publication once it meets all outstanding technical requirements.

Kind regards,

Ivan Filipe de Almeida Lopes Fernandes, Ph.D.

Academic Editor

PLOS ONE
---

## [Editor Report · Acceptance letter]

28 Dec 2023

PONE-D-23-10924R2 

PLOS ONE

Dear Dr. Monteiro dos Santos, 

I'm pleased to inform you that your manuscript has been deemed suitable for publication in PLOS ONE. Congratulations! Your manuscript is now being handed over to our production team.

Kind regards, 

on behalf of

Dr. Ivan Filipe de Almeida Lopes Fernandes 

Academic Editor

PLOS ONE